**Atmospherically-forced sea-level variability in western Hudson Bay, Canada**
Igor A. Dmitrenko[1,*], Denis L. Volkov[2,3], Tricia A. Stadnyk[4], Andrew Tefs[4], David G. Babb[1],
Sergey A. Kirillov[1], Alex Crawford[1], Kevin Sydor[5], and David G. Barber[1]
[1]Centre for Earth Observation Science, University of Manitoba, Winnipeg, Manitoba, Canada
[2]Cooperative Institute for Marine and Atmospheric Studies, University of Miami, Miami,
Florida, USA
[3]NOAA, Atlantic Oceanographic and Meteorological Laboratory, Miami, Florida, USA
[4]Department of Geography, University of Calgary, Calgary, Alberta, Canada
[5]Manitoba Hydro, Winnipeg, Manitoba, Canada
*Corresponding author, igor.dmitrenko@umanitoba.ca, 125 Dysart Rd., University of Manitoba,
Winnipeg, Manitoba R3T 2N2 Canada
**Abstract:** In recent years, significant trends toward earlier breakup and later freeze-up of sea-ice
in Hudson Bay have led to a considerable increase in shipping activity through the Port of
Churchill, which is located in western Hudson Bay and is the only deep-water ocean port in the
province of Manitoba. Therefore, understanding sea-level variability at the Port is an urgent issue
crucial for safe navigation and coastal infrastructure. Using tidal gauge data from the Port along
with an atmospheric reanalysis and Churchill River discharge, we assess environmental factors
impacting synoptic to seasonal variability of sea level at Churchill. An atmospheric vorticity
index used to describe the wind forcing was found to correlate with sea level at Churchill.
Statistical analyses show that, in contrast to earlier studies, local discharge from the Churchill
River can only explain up to 5% of the sea level variability. The cyclonic wind forcing
contributes from 22% during the ice-covered winter-spring season to 30% during the ice-free
summer-fall season due to cyclone-induced storm surge generated along the coast. Multiple
regression analysis revealed that wind forcing and local river discharge combined can explain up
to 32% of the sea level variability at Churchill. Our analysis further revealed that the seasonal
cycle of sea level at Churchill appears to be impacted by the seasonal cycle in atmospheric
circulation rather than by the seasonal cycle in local discharge from the Churchill River,
particularly post-construction of the Churchill River diversion in 1977. Sea level at Churchill
shows positive anomalies for September-November compared to June-August. This seasonal
difference was also revealed for the entire Hudson Bay coast using satellite-derived sea level
altimetry. This anomaly was associated with enhanced cyclonic atmospheric circulation during
fall, reaching a maximum in November, which forced storm surges along the coast. Complete
sea-ice cover during winter impedes momentum transfer from wind stress to the water column,
reducing the impact of wind forcing on sea level variability. Expanding our observations to the
bay-wide scale, we confirmed the process of wind-driven sea-level variability with (i) tidal-
gauge data from eastern Hudson Bay and (ii) satellite altimetry measurements. Ultimately, we
find that cyclonic winds generate sea level rise along the western and eastern coasts of Hudson
Bay at the synoptic and seasonal time scales, suggesting an amplification of the bay-wide
cyclonic geostrophic circulation in fall (October-November), when cyclonic vorticity is
enhanced, and Hudson Bay is ice-free.
**Keywords:**  Hudson Bay; sea level; Churchill River discharge; atmospheric vorticity.

**1. Introduction**
Hudson Bay in northeast Canada is a shallow (mean depth ~ 150 m), semi-enclosed sub-arctic
inland sea that is connected to the Labrador Sea through Hudson Strait (Figure 1). The Bay
occupies approximately 831,000 km$^2$, making it the world's largest inland sea, and is
characterized by a high annual volume of river discharge (712 km$^3$; *Déry et al*., 2005; 2011) and
a dynamic seasonal ice cover that exists from November/December to June/July (*Hochheim and
Barber*, 2010; 2014). The mean circulation in Hudson Bay is comprised of the wind-driven and
estuarine components, where the estuarine portion is driven by the riverine water input
(*Prinsenberg*, 1986a), and the wind-driven portion is attributed to prevailing along-shore winds
(e.g., *Ingram and Prinsenberg*, 1998; *Saucier et al*., 2004; *St-Laurent et al*., 2011; *Ridenour et*
*al*., 2019a; *Dmitrenko et al*., 2020). Model simulations by *Saucier et al*. (2004) show that the
cyclonic circulation is stronger during fall, reaching a maximum in November when the winds
are strongest, and weakest in spring when Hudson Bay has a complete sea-ice cover. *Dmitrenko*
*et al*. (2020), however, found that even during the ice covered season strong cyclones can
amplify water circulation in the Bay. This is consistent with conclusions by *St-Laurent et al*.
(2011), who noted that momentum is transmitted through the mobile ice pack to the water
column. The efficiency of momentum transmission through the mobile ice strongly depends on
sea-ice roughness, which is impacted by ice concentration and characteristic length scales of
roughness elements including pressure ridges, melt ponds etc. (e.g., *Lüpkes et al*., 2012;
*Tsamados et al*., 2014; *Joyce et al*., 2019). In particular, ice floes in a state of free drift within a
partial or weak ice cover, typical of the polynya area in western Hudson Bay, increase the
transfer of wind stress into the water column (*Schulze and Pickart*, 2012). Both velocity
measurements (*Prinsenberg*, 1986b; *Ingram and Prinsenberg*, 1998; *Dmitrenko et al*., 2020) and
model simulations (*Wang et al*., 1994*; Saucier et al*., 2004; *St-Laurent et al*., 2011; *Ridenour et*
*al*., 2019b) show that during summer, cyclonic water circulation produces a coastal transport
corridor that advects riverine water along the coast toward Hudson Strait and into the Labrador
Sea.
The local water mass of Hudson Bay is dominated by freshwater input comprised of river runoff
from the largest watershed in Canada and sea-ice meltwater (e.g., *Prinsenberg*, 1984, 1988,
1991; *Saucier and Dionne*, 1998; *Granskog et al*., 2009; *Eastwood et al*., 2020). The annual
mean discharge rate of $22.6 \times 10^3$ m$^3$ s$^{-1}$ corresponds to a net discharge of 712 km$^3$ of freshwater
per year (*Déry et al*., 2005, 2011). A similar volume of $742 \pm 10$ km$^3$ of freshwater is contained
within the ice pack by April (*Landy et al*., 2017). Freshwater transport in Hudson Bay exhibits a
strong seasonal cycle influenced by the timing of river discharge (e.g., *Déry et al*., 2005), the
annual melt/freeze cycle of sea ice (*Ingram and Prinsenberg*, 1998; *Saucier et al*., 2004; *Straneo*
*and Saucier*, 2008; *Granskog et al*., 2011), and seasonality of wind forcing (*Saucier et al*., 2004;
*St-Laurent et al*., 2011).
During the last decade, significant progress has been achieved in understanding the Hudson Bay
environmental system (e.g., *Granskog et al*., 2009; *Kuzyk et al*., 2011; *St-Laurent et al*., 2011;
*Piecuch and Ponte*, 2015; *Landy et al*., 2017; *Kuzyk and Candlish*, 2019; *Eastwood et al*., 2020;
*Dmitrenko et al*., 2020, 2021). However, the synoptic, seasonal, and interannual variability of sea
level in Hudson Bay still remains insufficiently studied due to a scarcity of sea level observations
at permanent tidal gauges. Note that the tidal gauge in Churchill (Figure 1) is the only
continuously operating tide gauge in Hudson Bay and the central Canadian Arctic. Historically,
the focus of sea level studies in Hudson Bay was motivated by this area's post-glacial isostatic
rebound (e.g., *Guttenberg*, 1941; *Tushingham*, 1992); for a detailed review of these earlier
studies see *Wolf et al*. (2006). The advent of space-geodesy, in particular GPS, absolute-
gravimetry, and satellite altimetry measurements (e.g., *Larson and van Dam*, 2000; *Wolf et al*.,
2006; *Sella et al*., 2007) afforded a shift in focus for Hudson Bay sea level research to
environmental aspects related to global warming and hydroelectric regulation (*Gough*, 1998,
2000), and those associated with increasing the shipping traffic from the Port of Churchill
through Hudson Bay to Hudson Strait, which may soon become a federally-designated
transportation corridor (e.g., *Andrews et al.*, 2017; *Pew Charitable Trusts*, 2016).
In 2016, the University of Manitoba and Manitoba Hydro launched a project on "Variability and
change of freshwater-marine coupling in the Hudson Bay System", named BaySys, which aimed
to assess the relative contributions of climate change and river regulation to the Hudson Bay
system. Here, we are specifically focused on the impact of the Churchill River diversion on
variability of sea level at the Port of Churchill. Additionally, we put our findings in the context of
wind forcing over the entire Hudson Bay, elaborating on the suggestion by *Dmitrenko et al.*
(2020) that cyclonic wind forcing generates onshore Ekman transport and storm surges along the
coast.
We also revisit earlier results by *Gough and Robinson* (2000) and *Gough et al.* (2005). Using
tidal gauge and river discharge data from 1974 to 1994, *Gough and Robinson* (2000) suggested
that the Churchill River discharge dominates sea-level variability at Churchill. They explained
the seasonal elevation of sea level during late fall by a recirculating mechanism that links the
spring pulse of river discharge in the downstream James Bay (Figure 1) to sea level at Churchill
(*Gough and Robinson*, 2000; *Gough et al.*, 2005). In this paper, we present an alternative
mechanism and show that (i) the Churchill River discharge plays a secondary role for generating
sea level anomalies at Churchill, and (ii) the synoptic and seasonal variability of sea level at
Churchill and over the entire Hudson Bay is impacted by the wind forcing described with an
atmospheric vorticity index (Figure 2).

**2. Data**

2.1. *Sea level*

The daily mean sea level data used in this study were retrieved from the Canadian Tides and
Water Levels Data Archive of the Fisheries and Oceans Canada through http://www.isdm-
gdsi.gc.ca/isdm-gdsi/twl-mne/index-eng.htm#s5 (last access: 26 August 2021). Sea level data
were de-tided using an algorithm by *Foreman* (1977). Measurements of sea level at Churchill
were obtained from the permanent tidal gauge that is installed at the port of Churchill (station
#5010) near the mouth of the Churchill River (Figure 1). While measurements of sea level at
Churchill date back to the 1930s (*Gutenberg*, 1941), we only used data from 1950 to present
(Figure 3a), which is coincident with atmospheric reanalysis data from the National Centers for
Environmental Prediction (NCEP; *Kalnay et al.*, 1996). In addition, we used sea level data from
the temporary tidal gauge in Innukjuak (station #4575), Cape Jones Island (station #4656), and
North Kopak Island (station #4548) (Figure 1). Among these three locations, only data at
Innukjuak are fully representative for our analysis because they span a sufficiently long period
from October 1969 to October 1980, however, only the portion of this time series from
September 1973 to December 1975 is continuous. Sea level records at Cape Jones Island and
North Kopak Island are from August-October 1973 and 1975, respectively, and were selected
among other temporary stations in Hudson Bay to overlap with sea level time series at
Innukjuak.
Satellite altimetry data from 1993-2020 were used to analyze the relationship between wind
forcing and sea level changes over the entire Hudson Bay. We used the daily fields of absolute
dynamic topography (ADT), i.e. the sea surface height above geoid, processed and distributed by
Copernicus Marine and Environment Monitoring Service (CMEMS;
https://marine.copernicus.eu/; last access: 26 August 2021). The ADT is obtained by adding a
mean dynamic topography (DT2018, *Mulet et al*., 2013) to sea level anomaly (SLA) measured
by altimetry satellites. The CMEMS SLA/ADT fields are computed by optimally interpolating
data from all satellites available at a given time following a methodology described in *Pujol et*
*al*. (2016). Prior to mapping, altimetry records are corrected for instrumental noise, orbit
determination error, atmospheric refraction, sea state bias, static and dynamic atmospheric
pressure effects, and tides. Because in this work we are interested in local (dynamic) changes of
sea level, the global mean sea level was subtracted from each ADT map. Then the seasonal
climatology was computed for June through August (JJA) and September through November
(SON) by averaging all available maps during the respective seasons. Sea ice does not represent
a significant problem for computing the climatology, because Hudson Bay is essentially ice free
during these months, especially during SON.
The root-mean-square differences between tide gauge records and collocated SLA/ADT data are
usually 3-5 cm (e.g., *Volkov et al*., 2007; *Pascual et al.*, 2009; *Volkov et al*., 2012) and do not
exceed 10 cm globally (*CLS-DOS*, 2016). When the altimetry data are averaged to produce the
seasonal climatology, the measurement error is greatly reduced (at least by an order of
magnitude for 28 years of altimetry record). It should be noted that altimetry errors near the coast
are greater than in the open ocean. This is due to land contamination within the radar footprint
and to the fact that the geophysical corrections applied to altimetry data are usually optimized for
the open ocean and not for the coastal zones. In classical altimetry products, however, a large
percentage of data within 10–15 km from the coast is deemed invalid and not used for generating
SLA/ADT maps (e.g., *The Climate Change Initiative Coastal Sea Level Team*, 2020).
Furthermore, satellite altimetry data was used here only for a qualitative assessment of the basin-
scale seasonal sea-level patterns in Hudson Bay. Therefore, the reduced quality of altimetry
retrievals near the coast is not expected to impact the conclusions of this study.
2.2. *River discharge*
Churchill River discharge data were obtained from *Déry et al*. (2016) and extended to 2019;
thus, we use a continuous record of daily mean discharge from 1960 to 2019 (Figure 4a and
supplementary material). The record was constructed from gauged observations above Red Head
Rapids (station #06FD001), which is located ~87 km from the Churchill River mouth and is the
most downstream hydrometric gauge along the Churchill River. When these data were not
available, we used upstream gauges (applying a drainage area correction) to fill significant gaps
in the time series (see *Déry et al*. 2005 for detailed methods). Data were adjusted by drainage
area (between the hydrometric gauge location and river outlet) and any significant tributary
inflows were added to represent discharge at the outlet of the Churchill River.
2.3. *Wind forcing*
Fields of sea level pressure (SLP) and 10-m wind velocity at 6-h intervals were derived from the
NCEP atmospheric reanalysis (https://psl.noaa.gov/data/composites/hour/; last access: 26 August
2021). We chose the NCEP reanalysis to extend the atmospheric forcing data back to 1950,
which covers the tide gauge record from Churchill, while a previous comparison of wind speeds
from NCEP and ERA5 (*Copernicus Climate Change Service*, 2017; *Hersbach et al.*, 2020) with
in situ observations from the Churchill weather station revealed an insignificant discrepancy
between the two reanalyses and meteorological observations (*Dmitrenko et al.*, 2020). However,
we used the ERA5 SLP data to validate atmospheric vorticity derived from NCEP as described
below in section 3. For simplicity, cyclones over the Hudson Bay area were manually tracked for
August-May 1969-1970 and 2003-2004 using the NCEP SLP fields, with the central position and
low SLP tabulated. The horizontal resolution of the NCEP-derived data is 2.5° of latitude and
longitude.
For the majority of tidal gauge data from 1950s, sea level at Churchill was recorded hourly. In
contrast, the Churchill River discharge from gauged observations above Red Head Rapids
(station #06FD001) is available daily. The NCEP data on SLP and 10-m wind are available at 6-
h intervals. To make these three time series comparable, we analyzed daily means.

## 3. Methods

For the 1950/60–2019 study period, a vorticity index was derived from the daily mean SLP
NCEP data to characterize the wind forcing and compare to the time series of sea level anomalies
(Figures 2a, 3a, and 4a). The vorticity index gives both the sign and magnitude of atmospheric
vorticity; it was first proposed by *Walsh et al.* (1996) and then successfully used for describing
atmospheric forcing over the Siberian shelves (*Dmitrenko et al.*, 2008a; 2008b) and Hudson Bay
(*Dmitrenko et al.*, 2020). The vorticity index is defined as the numerator of the finite difference
Laplacian of SLP for an area within a radius of 550 km centered at 60°N and 85°W in Hudson
Bay (Figure 1). A positive index corresponds to cyclonic atmospheric circulation that is typically
associated with northerly winds in western Hudson Bay, whereas a negative vorticity index
corresponds to anticyclonic atmospheric circulation characterized by southerly winds in western
Hudson Bay (Figure 2). *Dmitrenko et al.* (2020) examined the spatial uncertainty of atmospheric
vorticity estimated at 60°N, 85°W by computing vorticity for the 5-point stencils with a central
node shifted relative to 60°N, 85°W by approximately 280 km northward, eastward, southward,
and westward. Their results show that vorticity computed at 60°N, 85°W best describes major
cyclonic storms observed in 2016–2017.
The vorticity index used in this study does not fully explain the observed variability of
meridional wind in western Hudson Bay (Figure 2b), which is mainly responsible for generating
storm surge along the coast (*Dmitrenko et al.*, 2020). However, vorticity describes the intensity
of cyclonic wind forcing over the entire Bay impacting the basin-scale circulation and sea level
deformations along the entire coastline of Hudson Bay (*Dmitrenko et al.*, 2020). Thus, our
approach allowed us to extend our findings over the entire Bay. We also conducted a validation
comparing the NCEP-derived vorticity to that derived from the ERA5 SLP utilizing the Web-
Based Reanalysis Intercomparison Tools (https://psl.noaa.gov/cgi-bin/data/testdap/timeseries.pl;
last access: 26 August 2021) described by *Smith et al.* (2014). The comparison showed
insignificant differences between the two reanalyses: the NCEP-derived vorticity only slightly
exceeds that obtained from ERA5, while the correlation between the NCEP and ERA5-derived
vorticities is 0.96 (Figure 2a).
The Churchill River discharge time series (Figure 4a) was compiled as follows. First, no
significant gaps in Churchill River discharge record occurred on a daily basis. There were,
however, some missing discharge data between 1976 and 1995, with some gaps up to 3 months
(e.g., 1984, 1987). When data gaps occurred, then the upstream hydrometric gauge below Fidler
Lake (station #06FB001) was used to infill data, with streamflow data adjusted to account for the
difference in contributing area between Fidler Lake and the Churchill outlet, following the
procedure of *Déry et al.* (2005). When the upstream hydrometric data were also unavailable, a
secondary step was taken to infill data gaps. Missing data on a given day were infilled using the
day-of-year mean value of streamflow over the available period of record. This procedure
constructed a daily climatology of streamflow (i.e., mean annual hydrograph) based on the
availability of data over the period of record.
For the Churchill River, however, we constructed a separate climatology of daily streamflow for
the periods prior to and after flow diversion in 1977. Partial diversion began in 1976, allowing
less than the full capacity of discharge to be diverted into the Nelson River system, with full
operation beginning in 1977. We therefore designated 1977 as the first year when diversion
became operational.
It is also important to separate the pre- and post-regulation periods for the analysis of the
potential impact natural (pre-diversion) and regulated Churchill River discharge have on sea
level anomalies at Churchill. *Déry et al.* (2016) reported that the Churchill River diversion
caused a significant decline in the mean annual discharge from $37.0 \pm 4.2$ km$^3$ year$^{-1}$ pre-
diversion (1964–73) compared to post-diversion flows ($8.4 \pm 2.9$ and $9.6 \pm 4.4$ km$^3$ year$^{-1}$ for
1984–93 and 1994–2003, respectively). *Déry et al.* (2016) further revealed the coefficient of
variation (CV) of annual Churchill River discharge increased in inter-decadal CV post-diversion
(1984–2013; CV = 0.35–0.67) compared to pre-diversion records (1964–1973; CV = 0.11). Both
the decline in mean annual discharge and increase in discharge variability for the post-diversion
period necessitate separate analysis of the impact of river discharge on sea level variability due
to non-stationarity in the discharge record, which was implemented in our analysis.
The sea level record in Churchill is impacted by the post-glacial isostatic adjustment, with
present-day uplift in the Hudson Bay area of ~10 mm year$^{-1}$ (e.g., *Sella et al.*, 2007). Combining
satellite altimeter data with the Churchill tide-gauge data gives an uplift rate of about $9.0 \pm 0.8$
mm year$^{-1}$ (*Ray*, 2015). The crustal uplift is evident in the negative sea level trend at Churchill of
about the same magnitude (Figure 3a). To examine synoptic to seasonal variability of sea level at
Churchill, a polynomial fit was subtracted from the data (Figure 3a). The polynomial fit better
explains long-term variability of sea level at Churchill compared to the linear approximation,
with respective coefficients of determination ($R^2$) of 0.41 and 39. Thus, in our study we
examined the sea level anomalies (SLA) against the low-frequency trend conditioned by the
post-glacial isostatic adjustment. In addition, the inverse barometer contribution to the water
level record was removed using sea-level atmospheric pressure from the NCEP reanalysis. The
mean correction attributed to inverted barometer effect was –1.19 ± 8.72 cm.
We used multiple linear regression to estimate a partial contribution of the cyclonic wind forcing
and Churchill River discharge to SLA. In this context, multiple regression uses the least squares
method to calculate the value of SLA based on the two independent variables as the vorticity
index and Churchill River discharge.

## 4. Results

In this section, we examine the impact of wind forcing and local river discharge on sea level
variability at Churchill. We analyze (4.1) SLA at Churchill, (4.2) atmospheric vorticity over
Hudson Bay, (4.3) the Churchill River discharge, and (4.4) their correlations.
4.1. *Sea level*
The 30-day running mean of SLA at Churchill ranging from 0.39 m in October 1973 to –0.36 m
in April 1981 is dominated by the seasonal cycle (Figure 4a, blue line). In terms of the long-term
monthly mean, sea level shows a seasonal cycle with positive anomalies > 0.09 m from
September-November and negative anomalies of about –0.14 m from March-April (Figure 5a).
There is a substantial difference in the seasonal patterns of sea level between the pre- and post-
diversion periods. The long-term variability of sea level (Figure 3a) and SLA (Figure 4a) shows
no abrupt disruption with the introduction of the Churchill River diversion in 1977. However, the
seasonal cycle of SLA generated for pre- and post-diversion shows a characteristic difference in
the timing and magnitude of SLA (Figure 5a). First, for the natural seasonal cycle prior to 1977
(blue line in Figure 5a), SLA shows two seasonal peaks in June (~0.04 m; standard error of the
mean σ = ±0.01 cm) and November (~0.11 m, σ = ±0.02 cm). Post-diversion, SLA shows no
peak in June, but the magnitude of positive anomalies in September and October increased to >
0.08 m. This result is consistent with findings by *Gough and Robinson* (2000). In contrast to
summer, during February-May, the pre- and post-diversion magnitude of SLA decreased and
increased, respectively, by ≥ ±0.02 m relative to the long-term monthly mean (Figure 5a). The
standard deviation of the monthly mean values is up to 0.1 m (error bars in Figure 5a). The
seasonal pattern of SLA was partially disrupted in 1981-82 and 1987-88, and significantly
diminished in 1962-63 and 2016-17 (Figures 3a and 4a).
A closer look at the daily data reveals that the sea level seasonal maximum from October-
November is modulated by storm surges frequently observed during the late fall. For example, in
1969-70 and 2003-04 (highlighted with yellow shading on Figure 4), the seasonal cycle of sea
level (Figure 6, thick light blue line) was impacted by synoptic-scale events dominant during
October-November (Figure 6, blue line). These storm surges lasted from ~3 to 6 days and
correspond to positive anomalies of up to 0.5 m in the daily mean sea level (Figure 6b). In
contrast, from December to May, the number and magnitude of storm surges gradually decrease
(Figure 6).
4.2. *Wind forcing*
The vorticity index shows predominant cyclonic atmospheric circulation over Hudson Bay
(mostly positive values in Figure 3a, red line), which agrees with results presented by *Saucier et*
*al*. (2004) and *St-Laurent et al*. (2011). The strongest positive (cyclonic) vorticity is observed
from fall 1962 to winter 1963 (vorticity index exceeded 14 $s^{-1}$), while the strongest negative
(anticyclonic) atmospheric forcing (vorticity $< 4$ $s^{-1}$) is recorded during summer 1963 (Figure
3a). Overall, the alternation between monthly mean cyclonic and anticyclonic wind forcing is
mostly governed by the seasonal cycle in vorticity (Figure 5b). The monthly mean vorticity
increases from 4 $s^{-1}$ in September to ~8 $s^{-1}$ in November, and then gradually returns to ~4 $s^{-1}$ in
February (Figure 5b). During March-May and August, vorticity is relatively low ($< 2$ $s^{-1}$), and
only in June and July does vorticity change to weak anticyclonic (slightly negative) values
(Figure 5b). The seasonal cycle in atmospheric vorticity shows an insignificant difference pre-
and post-diversion. From May to August and in December, there is no difference between the
long-term monthly mean and monthly mean estimates for pre- and post-diversion (Figure 5b).
For other months, the difference does not exceed $\pm0.7$ $s^{-1}$.
The interannual variability of wind forcing is mainly attributed to year-to-year changes in the
cyclonic atmospheric circulation during fall-winter months. The seasonal amplitude of vorticity
is significantly diminished in 1953-54, 2001-02 and 2015-2016 when the seasonal mean vorticity
index for late fall to the beginning of winter did not exceed 8 $s^{-1}$ (black triangles in Figure 3a). In
contrast, during 1960-65, the vorticity seasonal cycle is amplified with the seasonal mean
vorticity index between late fall and early winter up to 28 $s^{-1}$ (green triangles in Figure 3a). The
standard deviation of the monthly mean vorticity shown by error bars in Figure 5b gradually
decreases from $\pm4.5$ $s^{-1}$ in December to $\pm2.8$ $s^{-1}$ in March-April.
Analysis of the daily vorticity time series sheds light on the origin of seasonality in vorticity.
Positive seasonal anomalies from September-December (Figures 3a and 5b) are partly attributed
to the occurrence of numerous vorticity peaks. For example, in 1969-70 and 2003-04
(highlighted with yellow shading in Figure 3), the seasonal enhancement of atmospheric vorticity
(Figure 6, thick pink line) was partially conditioned by synoptic-scale events recorded during
October-November 1969 and 2003 (Figure 6, red line). The strongest vorticity peaks were
observed on 18 October and 25 November 1969 ($>4$ $s^{-1}$; Figure 6a) and 15 October and 21
November 2003 ($>5$ $s^{-1}$; Figure 6b). The SLP spatial distribution reveals that each of these peaks
is attributable to a cyclone passing over Hudson Bay, with the center of low SLP located over the
central Hudson Bay on 18 October and 25 November 1969 (Figures 7a and 7b, respectively) and
15 October and 21 November 2003 (Figures 7c and 7d, respectively). The horizontal gradients of
SLP over western Hudson Bay ranged from 0.020 hPa $km^{-1}$ (25 November 1969; Figure 7b) to
0.035 hPa $km^{-1}$ (21 November 2003; Figure 7d). Overall, from 1 September to 31 December,
vorticity exceeded 2 $s^{-1}$ nine and 12 times in 1969 and 2003, respectively. In contrast, from 1
January to 30 April 1970 and 2004, vorticity exceeded 2 $s^{-1}$ only four and seven times,
respectively (Figure 6). This suggests that the seasonal cycle in atmospheric vorticity is partially
governed by the number and strength of cyclones passing over Hudson Bay.
4.3. *Local river discharge*
The time series of Churchill River discharge (Figure 4a) is dominated by (i) the introduction of
the flow diversion in 1977 and (ii) the seasonal hydrologic cycle. The mean discharge dropped
by about one-third from 1,190 m$^3$ s$^{-1}$ (1960-1976) to about 400 m$^3$ s$^{-1}$ following the diversion in
1977. At the same time, the standard deviation of the mean discharge increased from about ±300
to ±470 m$^3$ s$^{-1}$ following the diversion (Figure 4a). This is in line with results by *Déry et al.*
(2016). The mean annual timing of maximum river discharge during late spring to summer is not
significantly disrupted by the diversion (Figure 5c). The magnitude of the monthly mean
discharge pre- to post-diversion, however, reduces from about five-fold in March to about two-
and-a-half-fold in May-August (Figure 5c). After diversion, the standard deviation of the
monthly mean discharge doubles from May to October (Figure 5c). In contrast, from December
to April, the standard deviation of the monthly mean was not significantly impacted by the
diversion (Figure 5c).
4.4. *Sea level response* to *wind forcing and local river discharge*
Our data shows that SLA in Churchill, atmospheric vorticity over Hudson Bay, and Churchill
River discharge all show variability dominated by the seasonal cycle (Figures 3a, 4a, and 5). In
what follows, SLA at Churchill is first compared to the atmospheric vorticity, and then to the
Churchill River discharge, with a main focus on the seasonal cycle.
The correlation between the daily vorticity index and SLA from 1950-2019 and 1960-2019 is
0.48 and 0.47, respectively, with insignificant differences between correlations estimated for
periods pre- and post-diversion (0.49 and 0.47, respectively; Figure 3b and Table 1). For the ice-
free period from June to November, correlations for whole period, and pre- and post-diversion
increase to 0.54, 0.52 and 0.55 (Table 2), respectively, compared to 0.47, 0.49 and 0.47 for the
ice-covered period from December to May (Table 3). We test the difference between correlations
estimated for the ice-covered and ice-free seasons using the Fisher z-transformation (*Fisher*,
1921). Statistical assessment shows that the only differences between correlations estimated for
whole period and post-diversion are statistically significant at the 99% confidence level.
The relationship between vorticity and SLA changes significantly from one year to another. The
mean annual correlations in Figure 3b show these differences ranging from 0.18 in 1982 to 0.69
in 1991. During periods when the sea level seasonal cycle almost disappears (1981-82 and 1987-
88), the mean annual correlation drops to about 0.3 and 0.4, respectively (Figure 3b). When the
sea level seasonal cycle is diminished (1962-63 and 2016-17), a modest correlation of ~0.5 is
estimated (Figure 3b). For time periods enlarged in Figure 6, the annual mean correlation
significantly exceeds the long-term mean of 0.47, attaining 0.65 and 0.57 for 1969-70 and 2004-
05, respectively (Figure 3b). The direct linkage between vorticity and SLA is evident in Figure 6.
During September-November 1969 and 2003, all significant synoptic peaks in SLA are
consistent with those in atmospheric vorticity, including storm surges on 18 October and 25
November 1969 (Figure 6a) and on 15 October and 21 November 2003 (Figure 6b).
In contrast to atmospheric vorticity, the correlation between daily SLA and river discharge is
significantly smaller. Through the full record from 1960 to 2019, the correlation is 0.22, with an
insignificant difference between pre- and post-diversion (0.20 and 0.23, respectively, Figure 4b
and Table 1). For the ice-free period from June to November, correlations drop close to or below
the level of statistically significant values for the whole and pre-diversion periods (0.08 and 0.03,
respectively), and to 0.11 post-diversion (Table 2) compared to 0.21, 0.12 and 0.19 for the ice-
covered period from December to May (Table 3). Note that the difference between correlations
estimated for the ice-covered and ice-free seasons is statistically significant for only 1960-2019.
Similar to the linkage between vorticity and SLA, the relationship between river discharge and
SLA shows significant interannual variability. Correlations computed through the 365-day
moving window show negative to positive values ranging from -0.3 to 0.7 with about 15% of
estimates below the level of statistical significance (Figure 4b). Among all events when the
amplitude of the sea level seasonal cycle was strongly reduced, only 1962-63 and 1981-82 show
statistically significant correlation between river discharge and SLA of ~0.25 (Figure 4b). For
events in 1987-88 and 2016-17, correlation is relatively close to or below the level of statistical
significance (Figure 4b). The interannual difference in contribution of river discharge to the sea
level variability is also evident for 1969-70 and 2004-05. In 1969-70, the annual mean
correlation shows relatively modest contributions of river discharge to sea level variability
(correlation $R\sim0.29$; Figure 4b) as compared to correlation with atmospheric vorticity ($R\sim0.65$;
Figure 3b). In 2004-05, however, there is no correlation between SLA and river discharge
(Figure 4b), and sea level variability is impacted by wind forcing ($R = 0.57$; Figure 3b).
Overall, our results show that the wind forcing impacts the synoptic and seasonal variability of
sea level. In what follows, we use the coefficient of determination ($R^2$, where $R$ is correlation
coefficient in Tables 1-3) to describe the proportion of the variance in sea level that is explained
by the wind forcing, river discharge, and the wind forcing and river discharge together. Through
the whole annual cycle from 1960 to 2019, wind forcing explains about 22% of sea level
variability, while river discharge contributes only ~5%. Multiple regression analysis shows that
on average, both explain ~28% of sea level variability (Table 1).
Our results also reveal the important role of sea-ice cover and river diversion in modifying
controls on sea level variability. During the ice-free seasons from 1960-1976, the contribution of
wind forcing is 27%, and the role of river discharge is negligible (Table 2). Post-diversion,
cyclonic wind forcing and river discharge contribute 30% and 1%, respectively. Together they
explain up to 32% of sea level variability (Table 2). During the ice-covered season, the
contribution of vorticity is reduced to 22%, with insignificant difference between pre- and post-
diversion (Table 3). The contribution of river discharge varies from 1% for pre-diversion to 4%
for post-diversion. Wind and river forcing together explain ~27% of sea level variability for both
pre- and post-diversion periods (Table 3). Summarizing these results, we point out that the sea-
ice cover reduces the influence of wind forcing, and the influence of local river discharge is
slightly increased primarily during the ice covered post-diversion period. Post-diversion, the
magnitude of river discharge was reduced about three-fold, but seasonal variability increased by
a factor of 1.5 (Figure 4a and *Déry et al.*, 2016). Thus, we attribute the increase in river
discharge forcing during the post-diversion period mainly to the higher variability in river
discharge from May to November (Figure 4a, 5c, and *Déry et al.*, 2016). Note that during May
about 85% of Hudson Bay is ice covered (*Tivy et al.*, 2010), and the standard deviation of the
monthly mean discharge in May increases from about ±170 pre-diversion to ±380 m$^3$ s$^{-1}$ post-
diversion.

**5. Discussion**
Our results show that sea level variability at Churchill is rather influenced by wind forcing, with
discharge from the Churchill River playing a secondary role. Overall, the atmospheric vorticity
explains up to 30% of sea level variability at Churchill, with local river discharge contributing up
to only 5% (Tables 1-3). This suggests that in western Hudson Bay the northerly winds
associated with cyclonic wind forcing (Figure 2b) generate storm surge along the coast due to a
surface Ekman on-shore transport. This is consistent with results from *Dmitrenko et al*. (2020),
who used mooring records and Churchill tide gauge observations in 2016-17 to identify this
mechanism. A direct response of the water level to balance wind stress acting on the surface does
not play a role for generating SLA because there is no correlation between SLA and zonal wind
(not shown).
The SLA seasonal cycle in Figure 5a is only partially explained by seasonality in wind forcing
and local river discharge. The SLA seasonal cycle is also consistent with summertime warming
and freshening, and wintertime cooling and salinification. During the ice-free summer period, the
water column warms, and seawater becomes less dense and expands, causing the thermosteric
sea-level rise. In addition, during summer, riverine water and sea-ice meltwater decrease salinity
of the Bay, thus, causing the halosteric sea-level rise. It seems that these factors can explain the
significant fraction of the SLA seasonal variability that is not explained by wind forcing and
local river discharge. However, the detailed assessment of the thermosteric and halosteric
contributions to the Hudson Bay sea level variability is beyond the scope of this paper. In this
context, we point out that we examine only the direct impact of the river discharge on the sea
level in the Churchill River mouth ignoring the cumulative effect of riverine water on steric
height. This simplification seems to be reasonable because the residence time of the riverine
water fraction in southwestern Hudson Bay during summer is ~1-3 months (*Granskog et al*.,
461    2009).

For the seasonal time scales, increased cyclonic activity during fall to early winter impacts the
seasonal cycle in SLA. In contrast to *Gough and Robinson* (2000), we assert that a positive SLA
from September-November (Figure 5a) is attributed to enhanced atmospheric vorticity rather
than to the local river discharge. The signature of the local river discharge is, however, traceable
through the SLA seasonal cycle. During the pre-diversion period, positive SLA in June (Figure
5a) appears to be linked to the spring freshet of the Churchill River (Figures 5a and 5c).
However, post-diversion this positive SLA in June vanishes due to the abrupt decrease in the
Churchill River discharge during the spring freshet from ~1,500 to 700 m$^3$ s$^{-1}$ (Figure 5c).
Gradual decreases in Churchill River discharge from June/July to April for both pre- and post-
diversion cannot explain the positive SLA from fall to winter, especially during the post-
diversion period when the mean annual Churchill River discharge decreases to ~400 m$^3$ s$^{-1}$
(Figure 5c). Note that the cumulative effect of riverine water on steric height is neglected.
An additional perspective on SLA response to atmospheric and river forcing comes from a
comparison of the monthly mean vorticity and Churchill River discharge time series with SLA at
Churchill for the whole period of river discharge observations, and the pre- and post-diversion
periods (Figures 8a, 8b, and 8c, respectively). The SLA patterns for the whole period of river
discharge observation (Figure 8a) are strongly impacted by changes in the magnitude of
discharge during the pre- and post-diversion periods, as previously discussed. In contrast, the
SLA patterns compiled for the pre- and post-diversion periods (Figures 8b and 8c, respectively)
provide more precise features of the SLA response to atmospheric and river forcing. In general,
comparing atmospheric vorticity to sea level at Churchill shows that cyclones generate positive
SLA up to 0.15 m (Figure 8c). The maximum SLA response to cyclonic atmospheric forcing is
observed during the ice-free period (pink shading and white circles in Figures 8b and 8c), which
is consistent with results of the correlation analysis (Tables 2 and 3). The combination of
anticyclonic (negative) vorticity and low river discharge generates negative SLA up to 0.09 m
during both ice-free and ice-covered seasons (blue shading in Figures 8b and 8c).
The zero SLA contour in Figure 8b and 8c is displaced relative to the zero vorticity and the long-
term mean river discharge for the pre- and post-diversion periods. This indicates that these two
predictors alone are insufficient to entirely explain the sea level variability, and that there must
be other contributing factors. Correlation analysis (Tables 2 and 3) suggests that sea-ice also
plays a role in modifying the impact of atmospheric forcing on SLA. In this context, Figures 8
reveals the role of sea-ice cover for generating the SLA. The sea level at Churchill exhibits
negative SLA while atmospheric vorticity is positive, but not exceeding ~6-8 $s^{-1}$ (Figure 8). This
situation is usually observed during the ice-covered season when river discharge is below the
annual mean (blue circles and blue shading in Figures 8b and 8c). We attribute this disruption to
the sea-ice cover. Throughout the entire year, positive SLA is generated in response to strong
cyclones with vorticity exceeding ~6-8 $s^{-1}$ regardless of the river discharge contribution and sea-
ice conditions (red shading in Figure 8 for vorticity >~6-8 $s^{-1}$). During the ice-covered season, at
relatively low river discharge (<1,200 $m^3$ $s^{-1}$ and 350 $m^3$ $s^{-1}$ for pre- and post-diversion,
respectively), negative SLA is associated with positive vorticity <6-8 $s^{-1}$ (blue circles and blue
shading in Figures 8b and 8c). Thus, vorticity ~6-8 $s^{-1}$ is suggested to be a very rough estimate of
the vorticity threshold attributed to the sea-ice impact. Above this threshold, sea-ice does not
eliminate wind stress from the water column, and wind forcing impacts sea level variability in
Churchill year-round. Below this threshold, sea-ice eliminates wind forcing and a negative SLA
is conditioned by low river discharge. In fact, extension of the landfast ice as well as sea-ice
roughness and concentration can play a role modifying the thresholds at which wind impacts the
SLA. When the Churchill River discharge exceeds the monthly means of 1,500-1,600 $m^3$ $s^{-1}$ and
~900 $m^3$ $s^{-1}$ for pre- and post-diversion periods, respectively, positive SLA results regardless of
wind forcing.
Our results on the mechanisms of sea level variability at Churchill differ from those obtained by
*Gough and Robinson* (2000). First, using sea level and river discharge data from 1974-1994, they
found that correlation between Churchill River discharge and SLA in Churchill explains 43% of
sea level variability (versus the 5% derived in our analysis). Second, *Gough and Robinson*
(2000) explain a positive SLA observed in Churchill from October-November by the river
discharge pulse into the James Bay region with an advective lag of ~4-5 months. Furthermore,
*Gough et al.* (2005) speculate that positive SLA during fall is attributed to the James Bay
riverine water fraction, which does not exit the Bay through Hudson Strait, but instead re-
circulates in western Hudson Bay. The halosteric sea level changes associated with this
freshwater fraction are suggested to generate a positive SLA observed in Churchill from
October-November. The pathway of this water and the reason for disrupting the mean cyclonic
circulation in the Bay were, however, neither specified in *Gough and Robinson* (2000) nor in
*Gough et al.* (2005). The distance from James Bay to Churchill measured along the coast is
roughly 1,000 km. For a 120-150-day lag between peaks in river discharge to James Bay in June
(*Déry et al.*, 2005) and maximum positive SLA at Churchill in November, this distance suggests
the unrealistic rate of mean advective velocity to be ~8-10 cm s$^{-1}$. Note that *Dmitrenko et al.*
(2020) estimated the velocity of the northward flow along the western coast of Hudson Bay
during strong cyclonic storms to ~13 cm s$^{-1}$, which significantly exceeds the annual mean
meridional transport of ~1-2 cm s$^{-1}$.
Overall, the hypothesis by *Gough and Robinson* (2000) and *Gough et al.* (2005) about the
linkage between the river discharge pulse into James Bay and a positive SLA in Churchill is
suggestive of the seasonal disruption of the Hudson Bay cyclonic circulation that is in line with
the seasonal pattern of atmospheric vorticity in Figure 5b. Based on satellite altimetry and
numerical simulation, *Ridenour et al.* (2019a) revealed a seasonal reversal to anticyclonic
circulation in southwestern Hudson Bay from May-July, with a return to strong cyclonic
circulation in fall in response to the seasonal patterns of surface stress. This is consistent with the
seasonal cycles of vorticity presented in Figure 5b. However, among ~120-150 days of the
hypothetical transit time from James Bay to Churchill, the anticyclonic atmospheric forcing is
persistently observed only during May-July; in August, vorticity returns to cyclonic (Figure 5b).
In the three months before the occurrence of the positive SLA at Churchill in November, the
atmospheric forcing has already retuned to cyclonic (Figure 5b). In this context, the hypothesis
by *Gough and Robinson* (2000) and *Gough et al.* (2005) linking SLA in Churchill to river
discharge in James Bay seems to be inconsistent. In what follows, we provide additional
arguments to support our finding on the role of wind forcing in generating the SLA at Churchill.
First, *Tushingham* (1992) provide the time series of sea level at Churchill and the Churchill River
discharge from 1972 to 1989 (Figure 5 from *Tushingham*, 1992). These time series clearly show
an overall low positive correlation completely disrupted in 1973-74, 1977, and 1987-86, which is
consistent with our analysis (Figure 4). For 1973-74 and 1987-86, the annual-mean correlation
was estimated to be about –0.1 and is below the level of statistical significance (Figure 4b).
Overall, from 1960 to 2019, there were 19 events that lasted up to 1.8 years in duration when
correlations between the SLA and river discharge were statistically insignificant or even negative
(Figure 4b). This calls into question the correlations between Churchill River discharge and SLA
in Churchill reported by *Gough and Robinson* (2000) and *Gough et al.* (2005). Note that the
period from 1972 to 1989 used by *Tushingham* (1992) overlaps with the majority of the period
from 1974 to 1994 used by *Gough and Robinson* (2000).
Second, *Ward et al.* (2018) analyzed daily data from the Global Runoff Data Centre for 187
stations including Churchill and daily maxima sea level data from the Global Extreme Sea-level
Analysis. They found no statistically significant dependence between annual maxima of the
Churchill River discharge and sea level. For comparison, along the Pacific coast of North
America, the correlation ranged from 0.2 to 0.4, and accounted for 4-16% of the variation in sea
level. This is consistent with a previous concern about significant impact of Churchill River
discharge on SLA in Churchill.
Third, our analysis shows that the seasonal cycle in sea level variability with positive SLA
during fall is observed not only in Churchill, but also along the eastern coast of Hudson Bay in
Innukjuak (Figures 1 and 9). While the sea level record at Innukjuak is short and not continuous,
a positive SLA is recognizable during fall 1969-70 and 1973-76 (Figure 9, blue line). Note that
the seasonal SLA at Innukjuak cannot be generated locally because the annual mean (1964-2000)
discharge of the local Innuksuak River is only 3.3 km$^3$ year$^{-1}$, about three times smaller than the
Churchill River discharge post-diversion (*Godin et al.*, 2017). In contrast, the seasonal pattern in
SLA at Innukjuak is generated by the same cyclonic forcing as in Churchill. Seasonal SLA in
Innukjuak is consistent with seasonal amplification of atmospheric vorticity (Figures 5b and 9).
Moreover, in Innukjuak, the sea level peaks on 18 October and 25 November 1969 are coherent
with peaks in atmospheric vorticity (Figure 9) and sea level at Churchill (Figure 6a). From the
preceding analysis we explicitly know that these two vorticity peaks were generated by cyclones
passing over the Bay (Figure 7a). The coherent peaks in sea level in Churchill and Innukjuak
suggest that cyclones that were centered over Hudson Bay on 18 October and 25 November 1969
generated storm surge on both the eastern and western coasts of Hudson Bay. This is also
supported by a coherent response of sea level to atmospheric forcing at Cape Jones Island and
North Kopak Island (Figures 1 and 9). Our hypothesis is also consistent with results of sea level
numerical simulations in response to cyclones passing over the Bay in 2016-17 (*Dmitrenko et al.*,
2020). For synoptic storm surges, on-shore Ekman transport increases the mass of water column
along the coast (the barotropic component). The seasonal baroclinic component appears during
summer when water is fresher and warmer causing the thermosteric and halosteric sea-level rise
along the coast.
Fourth, satellite altimetry reveals a spatially uniform response of sea level to the seasonal cycle
in atmospheric vorticity along the whole coast of Hudson Bay (Figure 10). For 1993-2020, we
examine the difference between the sea surface heights (SSH) during summer, when monthly
mean atmospheric vorticity changes from –0.7 s$^{-1}$ in June to 1.1 s$^{-1}$ in August, and fall, when
vorticity increases from 4.2 s$^{-1}$ in September to 7.3 s$^{-1}$ in November (Figure 5b). Results suggest
that enhanced cyclonic vorticity during fall generates seasonal SSH elevation over the entire
coast of Hudson Bay with SSH differences between fall and summer ranging from >5 cm in
James Bay to ~1 cm along the northwest coast (Figure 10). This confirms our results that a
positive SLA during fall is generated over the entire coast of Hudson Bay, and particularly in
Churchill and Innukjuak, in response to enhanced cyclonic wind forcing (Figures 5a, 5b, and 9).
Overall, our third and fourth points suggest that the hypothesis of *Gough and Robinson* (2000)
and *Gough et al.* (2005) about a linkage between river discharge into James Bay and SLA in
Churchill is inconsistent.
One may suggest that seasonal SSH elevation in Figure 10 can be partly due to the thermosteric
and halosteric sea-level rise. During summer, the Hudson Bay coastal domain receives large
amount of fresh and warm water from river runoff. The seasonal tendency for river discharge,
however, is opposite to that for the SSH in Figure 10. For 1988-2000, *Déry et al.* (2005) reported
that the total discharge of rivers flowing into Hudson Bay peaks in June at ~3.6 km$^3$ day$^{-1}$, which
significantly exceeds the secondary maximum in October (~2.3 km$^3$ day$^{-1}$). The seasonal mean
total river discharge in September-November (~1.9 km$^3$ day$^{-1}$) is one-and-a-half times smaller
compared to ~2.8 km$^3$ day$^{-1}$ in June-August. Based on these estimates, the river discharge
seasonal cycle in June-November is inconsistent with that for the SSH in Figure 10. The
cumulative effect of river discharge on the seasonal cycle can play a role, but the residence time
of the riverine water fraction in southwestern Hudson Bay during summer is relatively small (~1-
3 months; *Granskog et al.*, 2009).
Finally, our results on the atmospheric forcing of the Hudson Bay SLA are in agreement with
conclusions by *Piecuch and Ponte* (2014, 2015). Using ocean mass measurements from satellite
gravimetry conducted during the Gravity Recovery and Climate Experiment, they found that
wind forcing dominates sea-level and mass variability in Hudson Bay, and wind might drive
Hudson Bay mass changes due to wind-driven outflow through Hudson Strait (*Piecuch and
Ponte*; 2014). For the sea level interannual variability in Hudson Bay, also evident in Figure 4a,
*Piecuch and Ponte* (2015) revealed a wind-driven barotropic fluctuation that explains most of the
non-seasonal sea level variance. Furthermore, they suggest that anomalous inflow and outflow
through Hudson Strait, which impacts sea level variability in Hudson Bay, are driven by wind
stress over Hudson Strait. This highlights the role of wind forcing in amplifying the freshwater
outflow from Hudson Bay, as also suggested by *Straneo and Saucier* (2008) and *Dmitrenko et al.*
621   (2020).

In summary, we suggest that seasonal amplification of atmospheric vorticity, partially
conditioned by the number and strength of cyclones passing over the Bay during fall to early
winter, generates the seasonal cycle in sea level variability over the entire Bay as depicted
schematically in Figure 11. Cyclones passing over Hudson Bay during fall to early winter cause
on-shore Ekman transport and storm surges over the entire coast of Hudson Bay (Figure 11a). In
summer, anticyclonic wind forces off-shore Ekman transport lowing sea level along the coastline
of Hudson Bay (Figure 11b).

**Summary and conclusions**

Our analysis revealed that in contrast to previous research, the local Churchill River discharge
explains only up to 5% of the sea level variability at Churchill. Cyclonic atmospheric forcing is
shown to explain from 22% during the ice-covered winter-spring season to 30% during the ice-
free summer-fall season (Tables 1-3). Multiple regression analysis showed that atmospheric
forcing and local river discharge together can explain up to 32% of the sea level variability at
Churchill. We found that a positive sea level anomaly in Churchill during fall is partially
conditioned by the seasonal cycle in atmospheric vorticity, with prevailing cyclonic wind forcing
during fall to the beginning of winter (Figure 5). Sea-ice cover reduces wind stress on the water
column during the ice-covered season from December to May, and cyclonic wind forcing
generates positive sea level anomalies at Churchill when only the monthly mean vorticity
exceeds ~6-8 s$^{-1}$ (Figure 8). In this context, transition towards a longer open water season (e.g.,
*Hochheim and Barber*, 2014) is expected to increase the contribution of atmospheric forcing to
sea level variability.
We expanded our observations at Churchill to the bay-wide scale using sea level observations
along the eastern coast of the Bay and satellite altimetry. A coherent sea level response to
atmospheric forcing observed at the opposite sides of Hudson Bay suggests that the spatial scale
of cyclones passing over Hudson Bay roughly equals the Hudson Bay area (Figures 7 and 9, and
*Dmitrenko et al.*, 2020). This scaling equivalency implies that cyclones passing over Hudson Bay
cause on-shore Ekman transport and storm surges over the entire Hudson Bay coast (Figure 11a).
This is also consistent with results by *Dmitrenko et al.* (2020) obtained for 2016-17. Moreover,
the satellite altimetry data shows that this scaling equivalency works not only for synoptic, but
also for the seasonal time scale. The seasonal cycle in atmospheric vorticity (Figure 5b) partially
conditions the seasonal cycle in sea level variability over the entire coast of Hudson Bay. The
recurring cyclonic wind forcing during fall favors sea level elevation over the entire Hudson Bay
coast compared to summer (Figures 10 and 11). This seasonal pattern in sea-level variability
seems to have implication for geostrophic circulation. The cross-shelf pressure gradient
generated due to seasonal amplification of sea level along the coast drives alongshore
geostrophic flow and favors the cyclonic circulation around Hudson Bay during fall to earlier
winter. In contrast, during summer the geostrophic component attributed to the anticyclonic
atmospheric forcing disrupts the Hudson Bay cyclonic circulation as shown by *Ridenour et al.*
(2019a).
Our research is important for maritime activity within the Bay. Communities around the Bay rely
heavily on the annual summer sea-lift to re-supply them at a fraction of the price compared to air
transport (*Kuzyk and Candlish*, 2019). In this context, positive coastal sea level anomalies during
fall favor re-supply operations to coastal communities. However, increased cyclonic activity
during fall is also associated with extreme wind events (Figure 2b) and storm surges (e.g., Figure
6) increasing risks to re-supply and fuel-transfer operations.
The origin of seasonality in wind forcing, its climatic aspects and ocean response to seasonal and
interannual variability in atmospheric vorticity over the Bay are among important priorities for
our future research. The freshwater storage in Hudson Bay and export through Hudson Strait
seem to be directly impacted by seasonal and interannual variability in wind forcing, clearly
defining the need for further research in this area using multi-year numerical simulations and
atmospheric reanalyses. Seasonality of the wind forcing is the hypothesized cause of the sea
level variability, but probably does not provide a complete explanation. The steric changes in
coastal zone attributed to river runoff were not taken into account that points out a necessity for
future research involving numerical simulations. Possible impacts of climate change on cyclone
activity in Hudson Bay, and therefore sea-level variability, will be addressed in future research.

**Data availability**

Sea level data used in this study are available from the Canadian Tides and Water Levels Data Archive of the Fisheries and Oceans Canada through http://www.isdm-gdsi.gc.ca/isdm-gdsi/twl-mne/index-eng.htm#s5 (last access: 26 August 2021). The daily SLA/ADT maps with all corrections applied are distributed via CMEMS (https://marine.copernicus.eu/; last access: 26 August 2021). Churchill River discharge data are provided in supplementary material. SLP and wind data are available from the https://psl.noaa.gov/data/composites/hour/ and https://psl.noaa.gov/cgi-bin/data/testdap/timeseries.pl (last access: 26 August 2021).

**Author contributions**

Conceptualization: ID; methodology: ID, DV, TS, AT; formal analysis: ID, DV, AT; investigation: ID, DV, AC, TS; resources: KS, DBarber; data curation: ID, DV, AT; writing (original draft): ID, DV, TS; writing (review & editing): AC, DV, SK, TS, AT, DBabb; visualization: ID, DV; supervision: DBarber; project administration: KS, DBarber; funding acquisition: KS, DBarber.

**Competing interests**

The authors declare that they have no conflict of interest.

**Acknowledgments**

This work is a part of research conducted under the framework of the Arctic Science Partnership (ASP) and ArcticNet. This research is also a contribution to the Natural Sciences and Engineering Council of Canada (NSERC) Collaborative Research and Development project: BaySys (CRDPJ470028-14). Funding for this work was provided by NSERC, Manitoba Hydro, the Canada Excellence Research Chair (CERC) program, the Canada Research Chairs (CRC) program and the Canada-150 Research Chairs program. D. Babb is additionally supported by NSERC and the Canadian Meteorological and Oceanographic Society (CMOS). DLV was supported by NOAA Atlantic Oceanographic and Meteorological Laboratory under the auspices of the Cooperative Institute for Marine and Atmospheric Studies (CIMAS), a cooperative institute of the University of Miami and NOAA, cooperative agreement NA20OAR4320472.

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

**Tables**
**Table 1:** Correlations (*R*) of daily atmospheric vorticity and/or Churchill River discharge against
sea level anomalies in western Hudson Bay for the whole annual cycle

| Predictor(s)/Time frame | 1960 - 2019 | Pre-diversion 1960 - 1976 | Post-diversion 1977 - 2019 |
|---|---|---|---|
| Vorticity | 0.47 | 0.49 | 0.47 |
| River discharge | 0.22 | 0.20 | 0.23 |
| Vorticity and river discharge* | 0.53* | 0.53* | 0.53* |


**Table 2:** Correlations (*R*) of monthly-mean atmospheric vorticity and/or Churchill River
discharge against sea level anomalies in western Hudson Bay for the ice-free period (June-
November)

| Predictor(s)/Time frame | 1960 - 2019 | Pre-diversion 1960 - 1976 | Post-diversion 1977 - 2019 |
|---|---|---|---|
| Vorticity | 0.54 | 0.52 | 0.55 |
| River discharge | 0.08 | 0.03** | 0.11 |
| Vorticity and river discharge* | 0.55* | 0.52* | 0.57* |


**Table 3:** Correlations (*R*) of monthly-mean atmospheric vorticity and/or Churchill River
discharge against sea level anomalies in western Hudson Bay for the ice-covered period
(December-May)

| Predictor(s)/Time frame | 1960 - 2019 | Pre-diversion 1960 - 1976 | Post-diversion 1977 - 2019 |
|---|---|---|---|
| Vorticity | 0.47 | 0.49 | 0.47 |
| River discharge | 0.21 | 0.12 | 0.19 |
| Vorticity and river discharge* | 0.52* | 0.51* | 0.52* |


**\*** The coefficient of multiple correlation is estimated based on the multiple linear regression
analysis
**\*\*** Correlation not statistically significant at the 99% confidence level
**Figures**


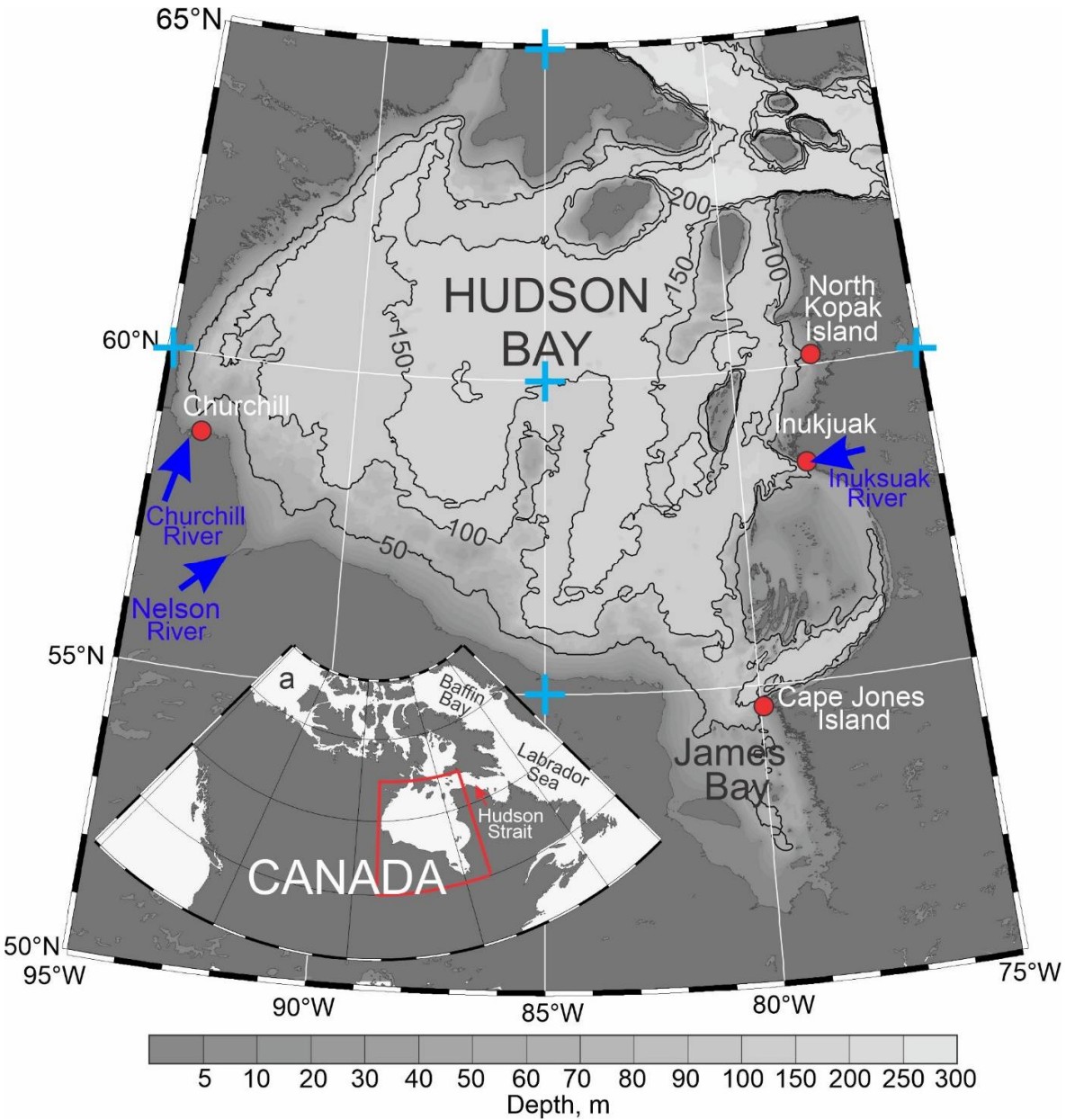


**Figure 1:** Map of Hudson Bay. Red dots depict the permanent tide gauge in Churchill and
temporary tide gauges in Innukjuak, Cape Jones Island and North Kopak Island. Blue arrows
highlight Churchill, Nelson and Inuksuak river mouths. Blue crosses depict the 5-point stencil
used for computing atmospheric vorticity approximated as Laplacian from sea level atmospheric
pressure. The numbered black lines depict depth contours of 50, 100, 150 and 200 m. (**a**) Inset
shows the Hudson Bay location within North America. The map of Hudson Bay was compiled
based on the General Bathymetric Chart of the Oceans (GEBCO, www.gebco.net).






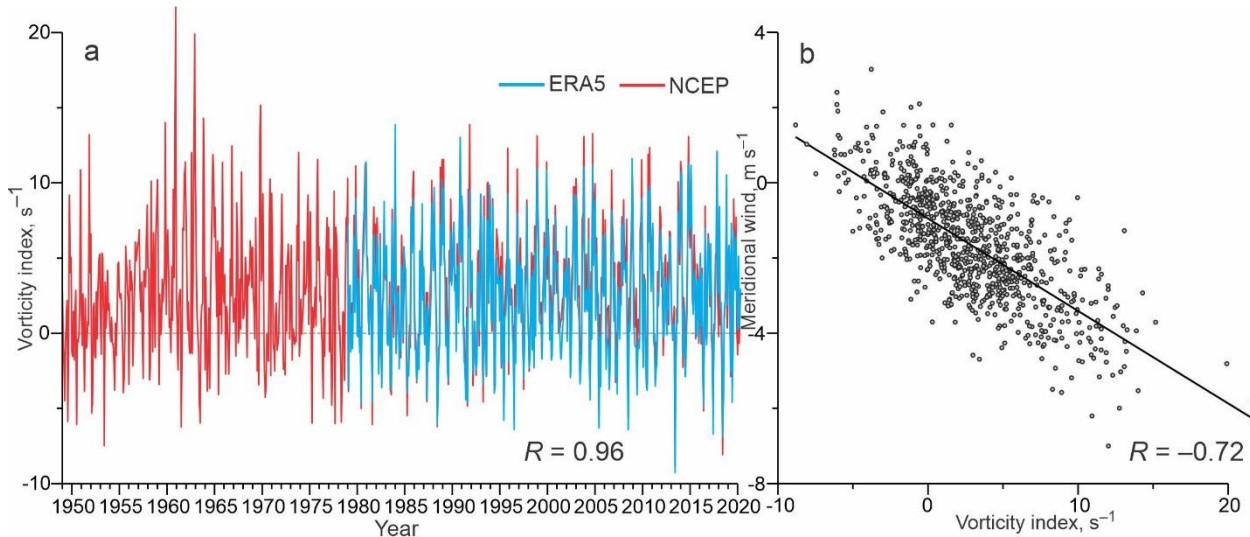


**Figure 2:** (**a**) Time series of the monthly mean atmospheric vorticity index (s⁻¹) over Hudson Bay, derived from NCEP (red) and ERA5 (blue). (**b**) Scatter plot of the monthly mean meridional wind seaward of Churchill in western Hudson Bay (m s⁻¹) versus the monthly mean atmospheric vorticity index. Thick black line depicts linear regression. Numbers at the bottom show correlation $R$ between (**a**) the monthly mean vorticity derived from NCEP (1949-2000) and ERA5 (1979-2000) and (**b**) the monthly mean NCEP vorticity versus meridional wind (1949-2020).

943

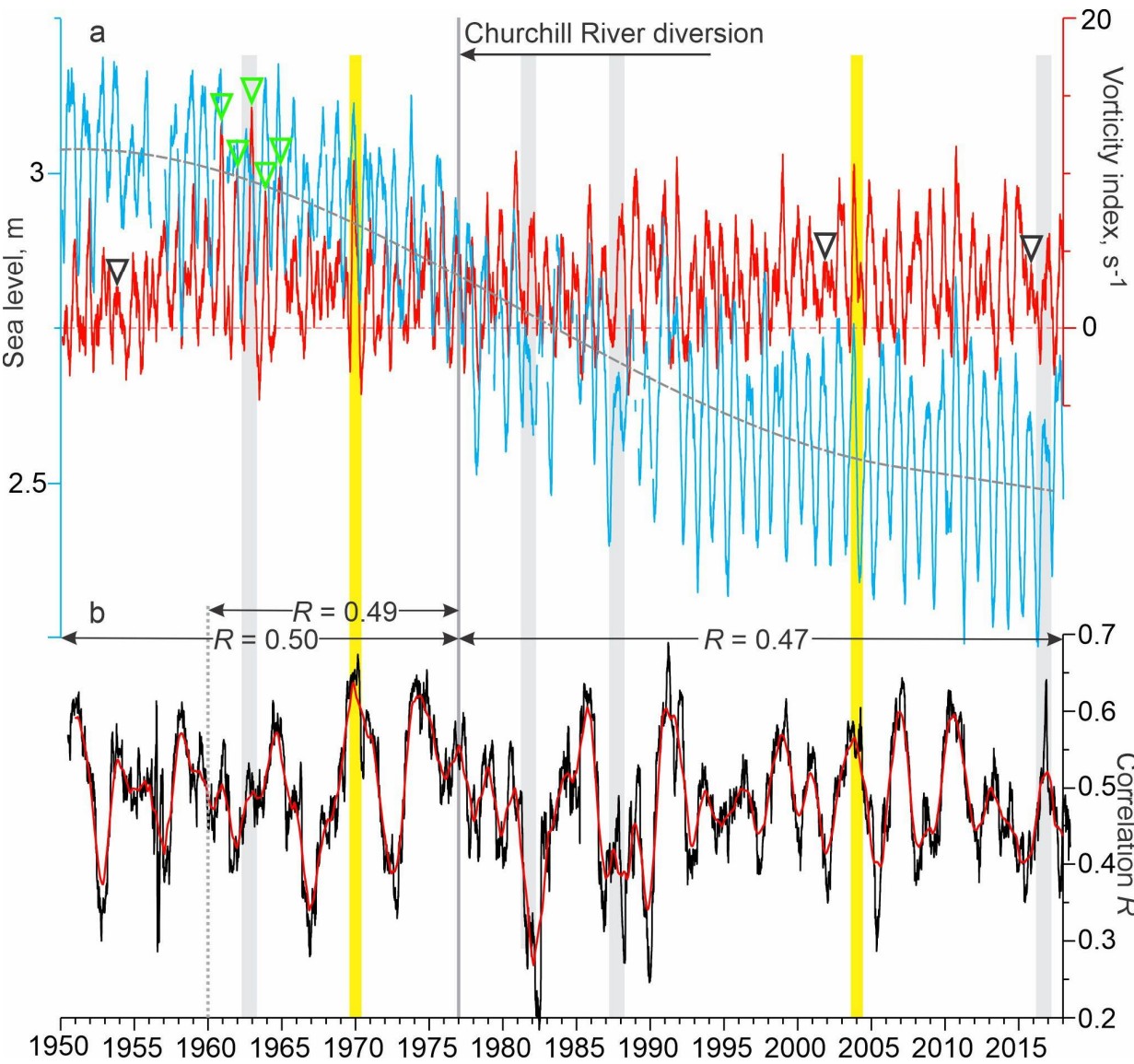

944

**Figure 3:** (**a**) 91-day running mean of daily mean atmospheric vorticity index (red, s$^{-1}$) over Hudson Bay and sea level measured at the tide gauge in Churchill (blue, m). Positive and negative vorticity correspond to cyclonic and anticyclonic atmospheric circulation, respectively. Gray dashed line shows polynomial approximation of the sea level trend attributed to the glacial isostatic adjustment. Black and green triangles show periods when seasonal vorticity from late fall to early winter was diminished and amplified, respectively. (**b**) Correlation $R$ between daily vorticity index and sea level anomaly (SLA) computed for the 365-day moving window (black) with their 365-day running mean (red). All correlations are statistically significant at 99% confidence. Numbers at the top show correlation between daily vorticity index and SLA computed for 1950/60-1976 and 1977-2018 pre- and post-diversion, respectively. (**a, b**) Yellow shading highlights August-May 1969-70 and 2003-04, enlarged in Figure 6. Black arrow indicates onset of the Churchill River diversion. Gray shading highlights periods when the sea level seasonal cycle was partially disrupted (1981-82 and 1987-88), or significantly diminished (1962-63 and 2016-2017).

959

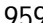

960

961

**Figure 4:** (**a**) 30-day running mean of the Churchill River discharge (black; $10^2$ m$^3$ s$^{-1}$) and detrended SLA at Churchill (blue; m). Gray circles show mean discharge pre- and post-diversion with standard deviations depicted with red error bars. (**b**) Correlation $R$ between daily Churchill River discharge and SLA computed for the 365-day moving window (black) with their 365-day running mean (red). Pink shading highlights statistically insignificant correlations at the 99% confidence level. Numbers at the top show correlation between daily Churchill River discharge and SLA computed for 1950-1976 and 1977-2018 pre- and post-diversion, respectively. (**a, b**) Yellow shading highlights August-May 1969-70 and 2003-04. Black arrow indicates onset of the Churchill River diversion. Gray shading highlights periods when the sea level seasonal cycle was partially disrupted (1981-82 and 1987-88), or significantly diminished (1962-63 and 2016-2017).

972

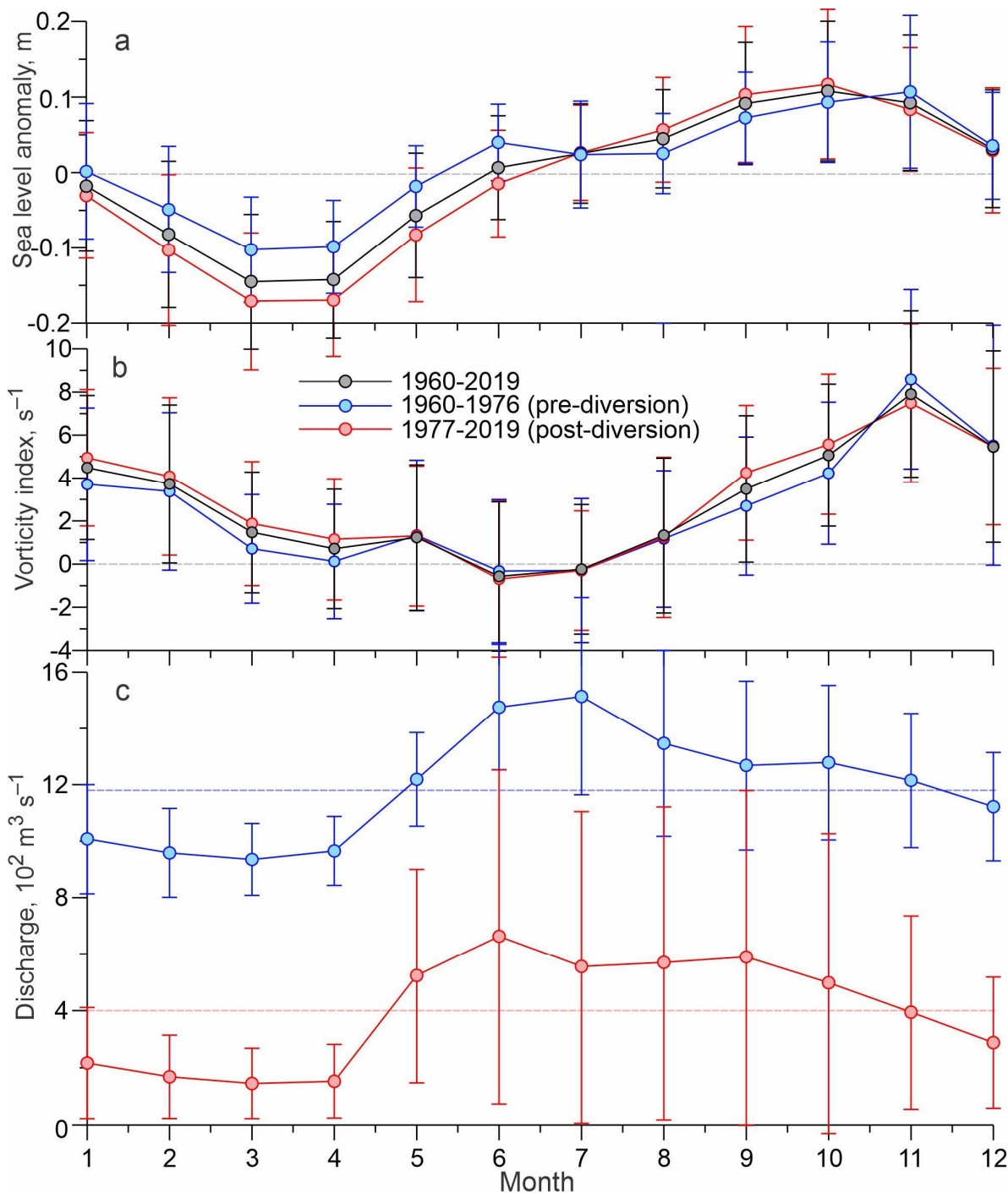

973

**Figure 5:** Seasonal cycle of (**a**) SLA at Churchill (m), (**b**) atmospheric vorticity over Hudson
Bay ($s^{-1}$), and (**c**) Churchill River discharge ($10^2$ $m^3$ $s^{-1}$). Seasonal cycle derived using monthly-
mean data for (**a, b**) 1950-2019 (black), (**a, b**) 1950-76 (blue) and (**c**) 1960-76 (blue) before the
Churchill River diversion, and (**a, b, c**) 1977-2018 (red) after the Churchill River diversion. Error
bars show ± one standard deviation of the mean. (**c**) Blue and pink dashed lines show the long-
term mean discharge before and after diversion, respectively.

980

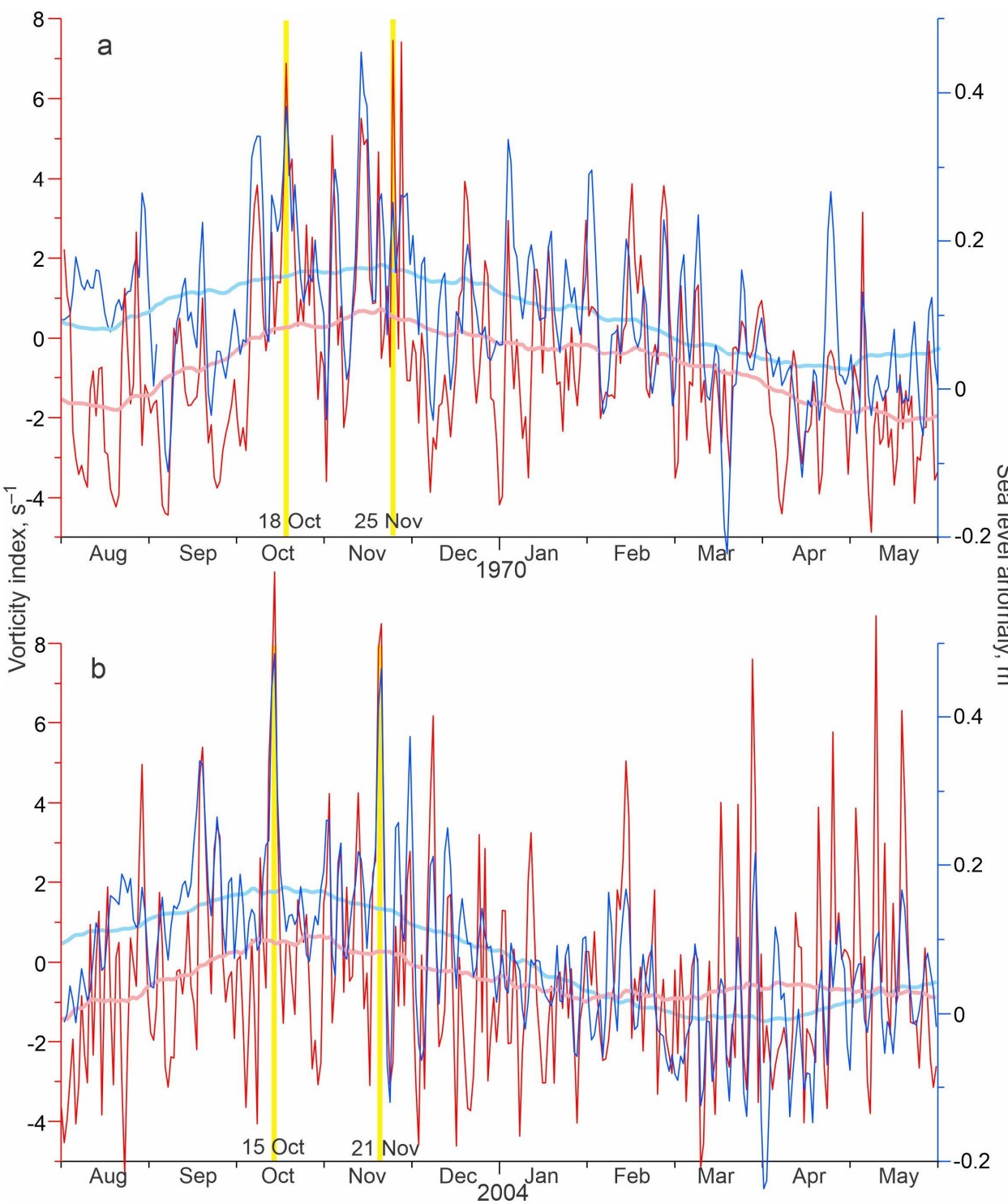

981

**Figure 6:** Time series of the daily mean vorticity index (red; s$^{-1}$) and SLA at Churchill (blue; m) with their 91-day running mean in pink and light blue, respectively, for August/May (**a**) 1969/1970 and (**b**) 2003/2004. (**a, b**) Vertical yellow lines highlight coherent peaks in vorticity and sea level in October and November.

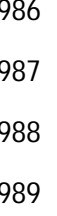






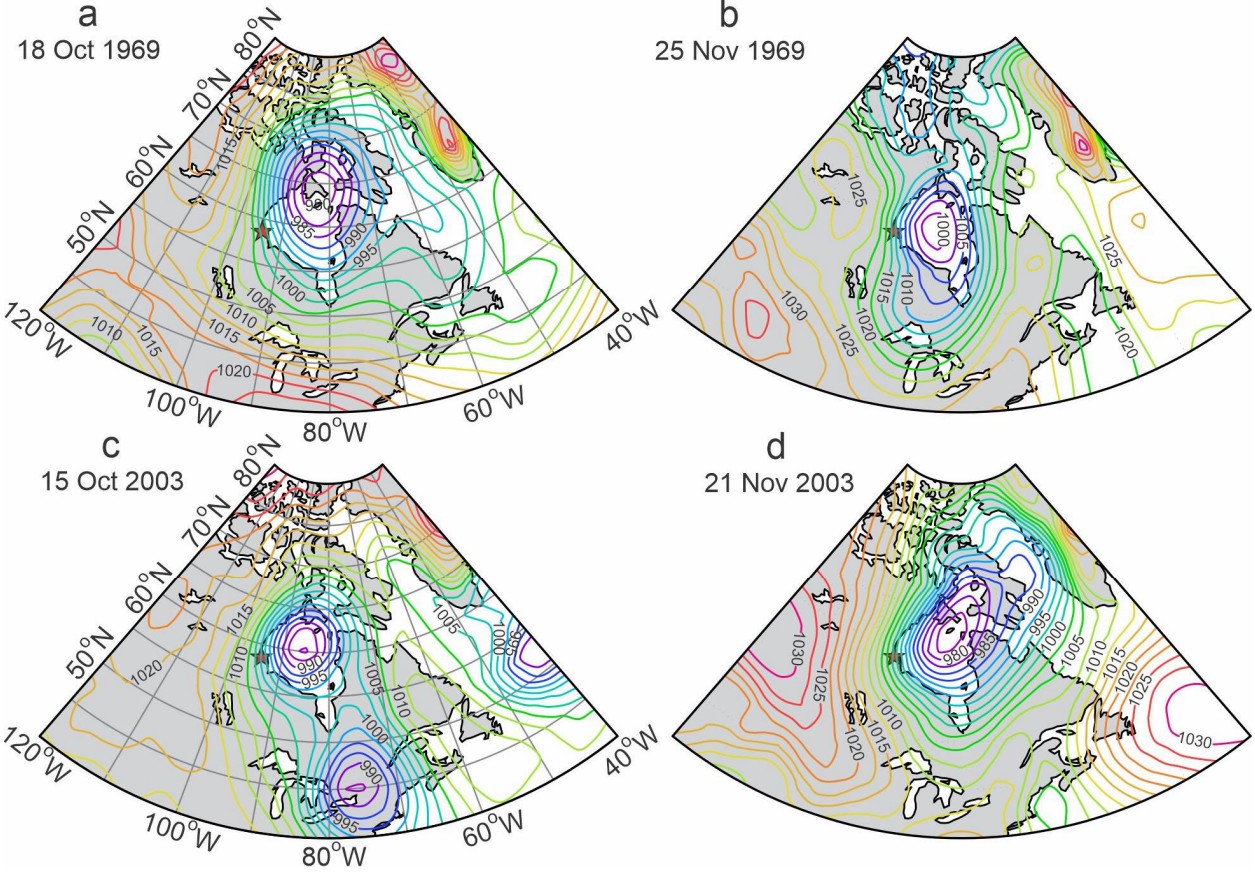


**Figure 7:** Sea level atmospheric pressure (hPa) for coherent peaks in atmospheric vorticity and sea level at Churchill, highlighted in Figure 6 with yellow lines: (**a**) 18 October 1969, (**b**) 25 November 1969, (**c**) 15 October 2003, and (**d**) 21 November 2003.









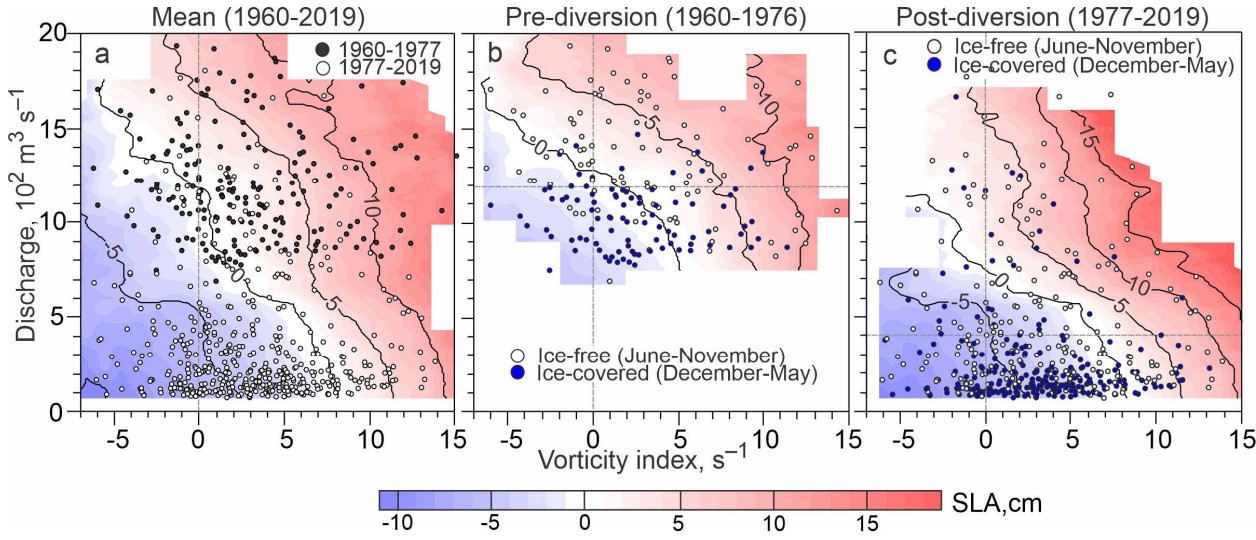

**Figure 8**: Color shading shows monthly mean sea level anomalies (cm) from tidal gauge at Churchill versus atmospheric vorticity (s$^{-1}$; horizontal axis) and Churchill River discharge ($10^2$ m$^3$ s$^{-1}$; vertical axis) for (**a**) entire period of river discharge observations (1960 – 2019), and (**b**) before and (**c**) after the Churchill River diversion in 1977. Scatter plots show monthly mean vorticity and river discharge for (**a**) 1960-1976 (black circles) and 1977-2019 (white circles), and (**b, c**) ice-free season (June-November; white circles) and ice-covered season (December-May; blue circles). Horizontal gray dashed line shows mean river discharge (**c**) before and (**d**) after diversion.









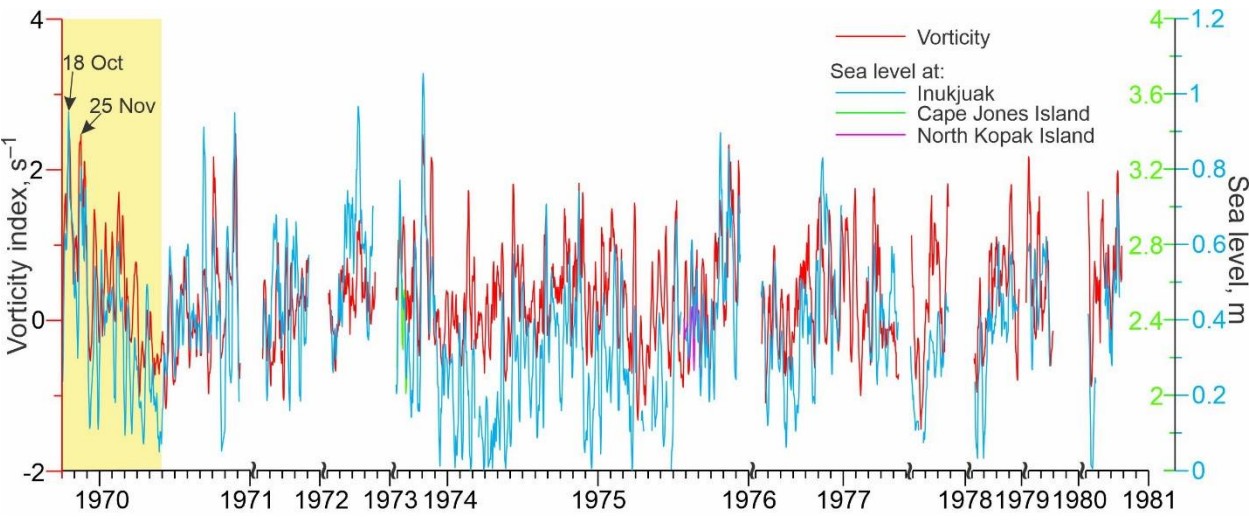


**Figure 9:** Time series of 7-day running mean for daily atmospheric vorticity index (red, s$^{-1}$) over Hudson Bay and daily mean sea level (m) measured at the tide gauge in Innukjuak (blue), Cape Jones Island (green) and North Kopak Island (purple). Yellow shading highlights October/May 1969/70. Black arrows indicate two cyclonic storms in 18 October and 25 November 1969 with atmospheric forcing shown in Figures 7a and 7b, respectively. Right vertical axis shows sea-level scale for Innukjuak (blue), and Cape Jones Island and North Kopak Island (green).






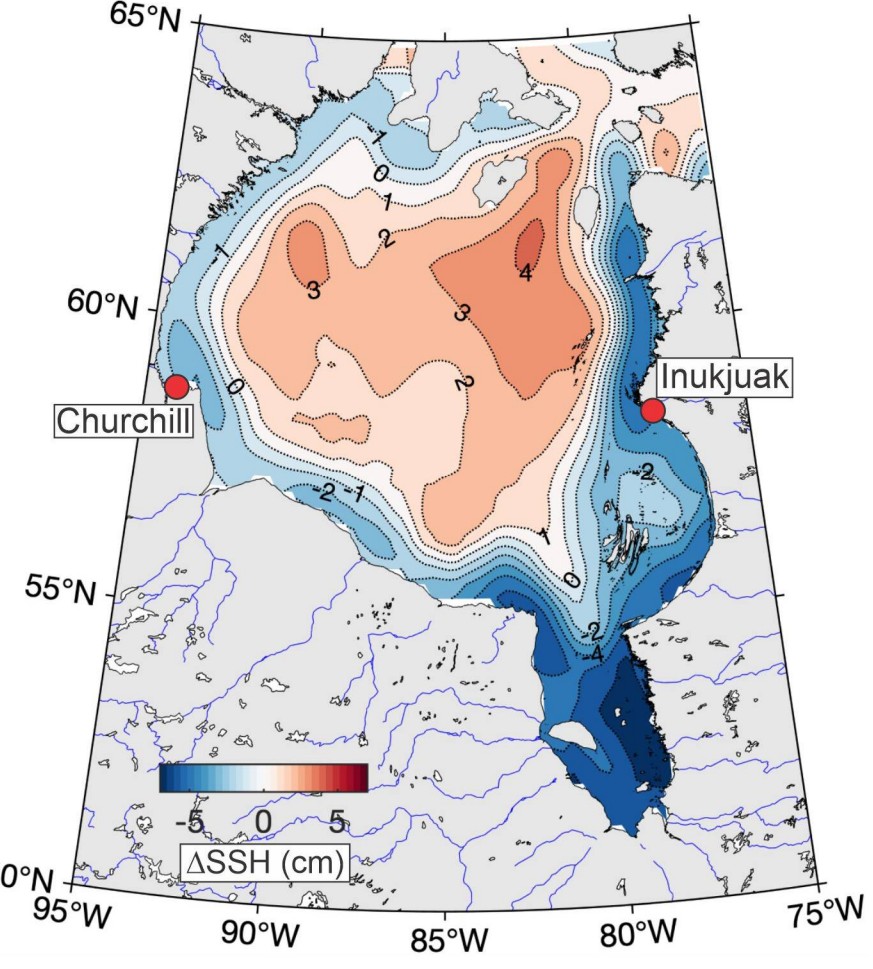


**Figure 10:** The long-term mean (1993-2020) difference between sea surface height (SSH; cm) in summer (June-August) and fall (September-November) derived from the satellite altimetry. Red dots depict the tide gauge in Churchill and Innukjuak.








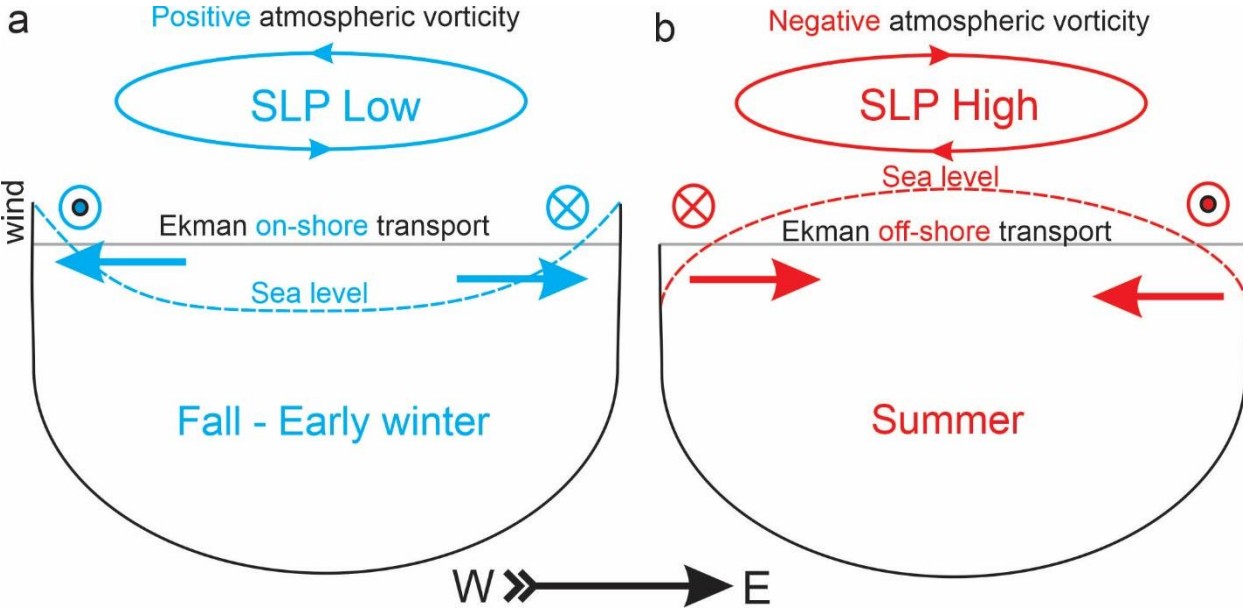


**Figure 11:** Diagram of the proposed impact of the seasonal changes in atmospheric vorticity on the sea level seasonal variability in Hudson Bay. (**a**) Positive (cyclonic) vorticity during October-December causes onshore Ekman transport and storm surges over the coast. (**b**) Negative (anticyclonic) vorticity during June-July forces offshore Ekman transport. During winter, a complete sea-ice cover reduces momentum transfer from wind stress to the water column diminishing impact of atmospheric forcing on sea level variability. Dotted and crossed circles depict southerly and northerly along-shore surface winds, respectively.

1051