# Peer review of "Atmospherically-forced sea-level variability in western Hudson Bay, Canada Igor A. Dmitrenko1,\*, Denis L. Volkov2,3, Tricia A. Stadnyk4, Andrew Tefs4, David G. Babb1, Sergey A. Kirillov1, Alex Crawford1, Kevin Sydor5<"

_Ocean Science, 2021_

## Referee Comment (RC1)

Title: Atmospherically-forced sea-level variability in western Hudson Bay, Canada
Author(s): Igor Dmitrenko et al.
MS No.: os-2021-50
MS type: Research article
Version: pre-print for public discussion, accessed 14 July 2021

**Referee comments:**

**Overall comments:**

With some clarifications, this is a useful contribution to the scientific body of knowledge surrounding processes affecting water levels in Hudson Bay, and provides new insight that atmospheric conditions play a more significant role in influencing water levels in this water body than previously thought. The piece is generally well-written, although could use a thorough final review to address a few minor grammatical, punctuation and sentence structure issues. I have noted a few examples in my specific comments where improvements could be made.

My overall impression is that the scientific analysis has been rigorous, although I have asked for a few clarifications on certain aspects in my specific comments, particularly surrounding the approach to inverse barometer correction, temporal averaging and de-trending of tide gauge data, importance of Ekman transport versus "conventional" storm surge generation processes, interpretation of satellite altimetry data and associated uncertainty, and the discussion of how sea ice affects SLA. With respect to the latter (role of sea ice), more nuance is needed, and the paper could benefit from referencing previous studies that have studied the role of sea ice conditions on atmosphere-sea momentum transfer and storm surges.

The quality of figures is generally quite good, with a few needs for additional annotation as identified in specific comments below. In general, the plots are a bit crowded (particularly Figures 3 and 4, which could use more separation between the upper and lower panels).

**Specific comments:**

Lines 16 and 18: suggest to check throughout for consistency in use of "sea-level" versus "sea level". Conventionally, a hyphen is only for use as a compound adjective, as in "sea-level rise". In this context, I would expect a reversal of the hyphen usage on lines 16 and 18, i.e. "variability of sea level" and "sea-level variability".

Line 38: Multiple uses of "cyclonic". Suggest to delete the first instance. In fact, I find the use of "cyclonic" is generally excessive throughout the paper, suggest to review and decide if really needed in all instances.

Lines 41-42: "even during the ice covered season strong cyclones can amplify cyclonic water circulation in the Bay". This seems to imply there is cyclonic circulation even in the absence of cyclones? Is it more correct to say that during periods of partial ice cover, the increased surface roughness and form drag imparted by ice floes to the water during strong cyclones can amplify cyclonic circulation compared to similar events coinciding with open-water conditions. Discussion of the role of sea ice regime on atmospheric momentum transfer to the water could benefit from referencing previous research in this area, e.g.:

- Lüpkes, C.; Gryanik, V.M.; Hartmann, J.; Andreas, E.L. (2012) A parametrization, based on sea ice morphology, of the neutral atmospheric drag coefficients for weather prediction and climate models. J. Geophys. Res. Space Phys., 117.
- Lüpkes, C. et al. (2013) Effect of sea ice morphology during Arctic summer on atmospheric drag coefficients used in climate models, JGR
- Tsamados et al. (2014) Impact of Variable Atmospheric and Oceanic Form Drag on Simulations of Arctic Sea Ice, American Meteorological Society
- Lupkes, C. et al. (2012) A parametrization, based on sea ice morphology, of the neutral atmospheric drag coefficients for weather prediction and climate models, JGR
- Andreas, E. et al. (2010) Parametrizing turbulent exchange over summer sea ice and the marginal ice zone
- Lu, P. et al. (2011) A parameterization of the ice-ocean drag coefficient, JGR.
- Shirasawa, K., Graham, R. (1991) Characteristics of the turbulent oceanic boundary layer under sea ice. Part 1: A review of the ice-ocean boundary layer, Journal of Marine Systems
- Hunke, E., and Dukowicz, J. (2002) The sea ice momentum equation in the free drift regime. Technical Report LA-UR-03-2219, Los Alamos National Laboratory, NM.
- Joyce, B.R.; Pringle,W.J.;Wirasaet, D.;Westerink, J.J.; Van DerWesthuysen, A.J.; Grumbine, R.; Feyen, J. High resolution modeling of western Alaskan tides and storm surge under varying sea ice conditions. Ocean Model. 2019, 141, 101421.
- Kim, J.; Murphy, E.; Nistor, I.; Ferguson, S.; Provan, M. Numerical Analysis of Storm Surges on Canada's Western Arctic Coastline. J. Mar. Sci. Eng. 2021, 9, 326.

Lines 85-86: I believe that water level data is available for the Churchill gauge at sub-daily (even sub-hourly) intervals. It would be helpful to comment on the implications of choosing to use the daily mean water level as the basis for evaluating SLA, given (i) a major focus of the study is on understanding the role of wind in contributing to storm surges, which are likely to manifest on time scales of the order of hours rather than days and (ii) likely disparities in SLA response time scales from wind and river discharge contributions, with the latter being more likely on the order of days to weeks.

Lines 113-114: The Déry et al. 2016 reference provided here refers the reader to another (2005) reference for details of the approach. I would suggest to directly reference the 2005 work, and provide quantitative values of parameters used in the drainage area correction to give the reader a feel for the proportion of total watershed for which this approach is used.

Line 171: The basis for the selection of polynomial fit to de-trend the data should be more clearly explained or justified, as well as the potential influence on the analysis as a whole. For example, the inflection point in the polynomial fit shown in Figure 3 coincidentally aligns quite closely on the time axis with the Churchill River Diversion. Is it possible that this de-trending method approach might de-emphasize the importance of Churchill River discharge contributions to SLA compared to, say, a linear de-trending approach? Could the selected approach also obscure other inter/intra-decadal influences on water level such as climate variability patterns? Are the correlations shown in Tables 1-3 sensitive to the choice of de-trending method?

Lines 172-173: It is not clear if the inverse barometer correction referred to here is a correction applied to the tide gauge record (in which case it should be clarified that the raw data represented measurements using an unvented tide gauge transducer) or simply a removal of the inverse barometer

(IB) contribution to the water level record. If the latter, it would be worth commenting on the magnitude of the contribution of IB to interannual SLA, given studies indicating it is often non-negligible (e.g. Piecuch & Ponte 2015) and the potentially significant correlation with vorticity/wind effects.

Lines 299-301: There have likely been changes in sea ice cover over the reference period. How is this reflected in the analysis, if at all? Given the dominance of wind effects under ice-free conditions, perhaps there is value in discussing potential future changes if, as expected, ice cover continues to decline.

Line 308: Is it correct to say that sea level variability at Churchill is *primarily* impacted by wind forcing, when the authors state that wind forcing only explains ~22% of variability (Line 290)?

Lines 310-312: Is there evidence to support the hypothesis that the dominant driver of storm surge on the western shore is Ekman transport, as opposed to a direct response of the water level to balance wind stress acting on the surface, or local bathymetric influences on storm surges? Although Figure 2b shows a strong correlation between the northerly wind component and vorticity, it would be helpful to more clearly explain whether these conditions are typically associated with an onshore (easterly) wind component or not, to confirm the relative importance of Ekman transport versus wind stress.

Lines 318-319: "It seems that these factors can explain the residual fraction of the SLA seasonal variability that is not explained by wind forcing and local river discharge." If I understand correctly, the authors concluded (Line 292) that wind effects and river discharge explain ~28% of SLA variability. The authors are therefore hypothesizing that the remainder, which at ~72% would represent a substantial (even dominant) contribution, is explained by thermosteric and halosteric effects. Is it then appropriate to refer to these as the "residual fraction"?

Lines 357-360: The vorticity threshold described appears to imply that sea ice conditions (i.e., concentration, thickness, roughness, mobility as opposed to simply presence/absence) have little or no influence on the role of wind forcing in explaining SLA variability. I would expect ice conditions (e.g., presence or absence of shorefast ice and/or dynamic ice, roughness, concentration, fraction of Hudson Bay covered by ice) to play a role in determining the thresholds at which wind influences (positively or negatively) the SLA.

Lines 377-378 and 385: Which hypothesis by Gough and Robinson (2000) and Gough et al. (2005) are the authors referring to in this section? The preceding paragraph identifies multiple distinct hypotheses, and the linkage is not clear. Both paragraphs would benefit from more clearly identifying which of G&R (2000) and G et al (2005) hypotheses are debunked, and which are simply subject to uncertainty.

Lines 417-418 (and 98-108): Figure 10 shows continuous contours of SSH derived from satellite altimetry data, which are discrete measurements at discrete intervals, usually along widely space satellite tracks. Altimetry data is also notoriously uncertain near the land/water interface. It would be helpful to provide some additional context on the number of observations, spatial and temporal intervals, how the contours were developed, and the uncertainty associated with the SSH measurements (considering absolute differences shown are no more than +/- 5cm). To what extent is the satellite altimetry analysis influenced by spatial and temporal resolution of data and uncertainty in interpreted sea level measurements in proximity to the land/water interface?

Lines 439-440: To what extent might anomalous inflows/outflows be controlled by the spatial distribution of sea ice (also potentially influenced by prevailing wind conditions)?

Lines 450-451: Odd use of "While…" to begin the sentence. Perhaps replace with "On the contrary".

Lines 466-470: I'm not sure the argument regarding the link between geostrophic flow and SLA is sufficiently put to rest without some analysis of the shore-perpendicular component of wind.

Lines 760-765: Symbols (concentric circles and crosses) require legend or definition within caption.

---

## Author Comment (AC1)

We highly appreciate helpful comments and suggestions by Reviewer #1. In the following, the comments by Reviewer #1 are underlined and our responses to the comments are in normal characters. Modifications to the text are shown in quotation marks with bold characters indicating newly added text, and normal characters indicating text that was already present in the previous version. The line numbering is referenced to the original version of our manuscript.

On behalf of the authors,

Igor A. Dmitrenko

**Reviewer #1**

Overall comments:

1. With some clarifications, this is a useful contribution to the scientific body of knowledge surrounding processes affecting water levels in Hudson Bay, and provides new insight that atmospheric conditions play a more significant role in influencing water levels in this water body than previously thought. The piece is generally well-written, although could use a thorough final review to address a few minor grammatical, punctuation and sentence structure issues. I have noted a few examples in my specific comments where improvements could be made.

We appreciate this favorable evaluation of our manuscript by Reviewer #1.

2. My overall impression is that the scientific analysis has been rigorous, although I have asked for a few clarifications on certain aspects in my specific comments, particularly surrounding the approach to inverse barometer correction, temporal averaging and de-trending of tide gauge data, importance of Ekman transport versus "conventional" storm surge generation processes, interpretation of satellite altimetry data and associated uncertainty, and the discussion of how sea ice affects SLA. With respect to the latter (role of sea ice), more nuance is needed, and the paper could benefit from referencing previous studies that have studied the role of sea ice conditions on atmosphere-sea momentum transfer and storm surges.

All these suggestions by Reviewer #1 were implemented in the revised version of manuscript.

3. The quality of figures is generally quite good, with a few needs for additional annotation as identified in specific comments below. In general, the plots are a bit crowded (particularly Figures 3 and 4, which could use more separation between the upper and lower panels).

We implemented more space between the upper and lower panels in Figures 3 and 4.

Specific comments:

4. Lines 16 and 18: suggest to check throughout for consistency in use of "sea-level" versus "sea level". Conventionally, a hyphen is only for use as a compound adjective, as in "sea-level rise". In this context, I would expect a reversal of the hyphen usage on lines 16 and 18, i.e. "variability of sea level" and "sea-level variability".

Modified as recommended.

5. Line 38: Multiple uses of "cyclonic". Suggest to delete the first instance. In fact, I find the use of "cyclonic" is generally excessive throughout the paper, suggest to review and decide if really needed in all instances.

We agree with this comment by Reviewer #1. We modified this sentence (line 38) as follows: "*Prevailing **along-shore** winds drive cyclonic circulation of water within the Bay...*". We also omitted "*cyclonic*" in several places below.

6. Lines 41-42: "even during the ice covered season strong cyclones can amplify cyclonic water circulation in the Bay". This seems to imply there is cyclonic circulation even in the absence of cyclones? Is it more correct to say that during periods of partial ice cover, the increased surface roughness and form drag imparted by ice floes to the water during strong cyclones can amplify cyclonic circulation compared to similar events coinciding with open-water conditions. Discussion of the role of sea ice regime on atmospheric momentum transfer to the water could benefit from referencing previous research in this area, e.g.:

Lüpkes, C.; Gryanik, V.M.; Hartmann, J.; Andreas, E.L. (2012) A parametrization, based on sea ice morphology, of the neutral atmospheric drag coefficients for weather prediction and climate models. J. Geophys. Res. Space Phys., 117.

Lüpkes, C. et al. (2013) Effect of sea ice morphology during Arctic summer on atmospheric drag coefficients used in climate models, JGR

Tsamados et al. (2014) Impact of Variable Atmospheric and Oceanic Form Drag on Simulations of Arctic Sea Ice, American Meteorological Society

Lupkes, C. et al. (2012) A parametrization, based on sea ice morphology, of the neutral atmospheric drag coefficients for weather prediction and climate models, JGR

Andreas, E. et al. (2010) Parametrizing turbulent exchange over summer sea ice and the marginal ice zone

Lu, P. et al. (2011) A parameterization of the ice-ocean drag coefficient, JGR.

Shirasawa, K., Graham, R. (1991) Characteristics of the turbulent oceanic boundary layer under sea ice. Part 1: A review of the ice-ocean boundary layer, Journal of Marine Systems

Hunke, E., and Dukowicz, J. (2002) The sea ice momentum equation in the free drift regime. Technical Report LA-UR-03-2219, Los Alamos National Laboratory, NM.

Joyce, B.R.; Pringle,W.J.;Wirasaet, D.;Westerink, J.J.; Van DerWesthuysen, A.J.; Grumbine, R.; Feyen, J. High resolution modeling of western Alaskan tides and storm surge under varying sea ice conditions. Ocean Model. 2019, 141, 101421.

Kim, J.; Murphy, E.; Nistor, I.; Ferguson, S.; Provan, M. Numerical Analysis of Storm Surges on Canada's Western Arctic Coastline. J. Mar. Sci. Eng. 2021, 9, 326.

Reviewer #1 correctly interpreted our text "*even during the ice covered season strong cyclones can amplify cyclonic water circulation in the Bay*". The Hudson Bay cyclonic circulation is observed even in the absence of cyclonic atmospheric forcing. ***The mean circulation in Hudson Bay is comprised of the***

*wind-driven and estuarine components, where the estuarine portion is driven by the riverine water input (Prinsenberg, 1986a), and the wind-driven portion is attributed to* prevailing along-shore winds *(e.g., Ingram and Prinsenberg, 1998; Saucier et al., 2004; St-Laurent et al., 2011; Ridenour et al., 2019a; Dmitrenko et al., 2020).* We added this text after line 37. Following suggestion by Reviewer #1, we also briefly introduced the role of sea ice on atmospheric momentum transfer to the water adding new text following line 44: "***The efficiency of momentum transmission through the mobile ice strongly depends on sea-ice roughness, which is impacted by ice concentration and characteristic length scales of roughness elements including pressure ridges, melt ponds etc. (e.g., Lüpkes et al., 2012; Tsamados et al., 2014; Joyce et al., 2019). In particular, ice floes in a state of free drift within a partial or weak ice cover, typical of the polynya area in western Hudson Bay, increase the transfer of wind stress into the water column (Schulze and Pickart, 2012)***". Reference list was updated accordingly.

7. Lines 85-86: I believe that water level data is available for the Churchill gauge at sub-daily (even sub-hourly) intervals. It would be helpful to comment on the implications of choosing to use the daily mean water level as the basis for evaluating SLA, given (i) a major focus of the study is on understanding the role of wind in contributing to storm surges, which are likely to manifest on time scales of the order of hours rather than days and (ii) likely disparities in SLA response time scales from wind and river discharge contributions, with the latter being more likely on the order of days to weeks.

For the majority of tidal gauge data from 1950s, sea level at Churchill was recorded hourly. In contrast, the Churchill River discharge data from gauged observations above Red Head Rapids (station #06FD001) are available at daily basis. The NCEP data on SLP and 10-m wind are available at 6-h intervals. To make these three time series comparable, we decided to analyze them as the daily mean data.  We added this statement following line 108 at the end of the data section: "***For the majority of tidal gauge data from 1950s, sea level at Churchill was recorded hourly. In contrast, the Churchill River discharge from gauged observations above Red Head Rapids is available daily. The NCEP data on SLP and 10-m wind are available at 6-h intervals. To make these three time series comparable, we analyzed daily means***".

8. Lines 113-114: The Déry et al. 2016 reference provided here refers the reader to another (2005) reference for details of the approach. I would suggest to directly reference the 2005 work, and provide quantitative values of parameters used in the drainage area correction to give the reader a feel for the proportion of total watershed for which this approach is used.

Reviewer #1 is correct. We changed this reference to *Déry et al.* (2005). However, providing details of the draining area corrections, particularly quantitative values of parameters used for the drainage area correction, seems to be excessive in this context.

9. Line 171: The basis for the selection of polynomial fit to de-trend the data should be more clearly explained or justified, as well as the potential influence on the analysis as a whole. For example, the inflection point in the polynomial fit shown in Figure 3 coincidentally aligns quite closely on the time axis with the Churchill River Diversion. Is it possible that this de-trending method approach might de-emphasize the importance of Churchill River discharge contributions to SLA compared to, say, a linear de-trending approach? Could the selected approach also obscure other inter/intra-decadal influences on water level such as climate variability patterns? Are the correlations shown in Tables 1-3 sensitive to the choice of de-trending method?

We generated figure (below) showing sea level (90-day running mean) and its linear and polynomial approximations. Figure shows that the linier fit slightly underestimates the sea level values before diversion and overestimates them after diversion. In this research, however, we are mainly interested in the sea level seasonal and synoptic variability. Thus, we are not focused on inter/intra-decadal influences on water level such as climate variability patterns. Because the sea-level time series is dominated by seasonal and synoptic variability, the correlations shown in Tables 1-3 are not sensitive to the choice of de-trending method.

[Figure]

For the linear approximation, the coefficient of determination ($R^2$) is 0.39. For the polynomial approximation $R^2$ = 0.41. This confirms that the polynomial approximation provides a better fit for the long-term trend attributed to the post-glacial isostatic adjustment. This is also obvious from the figure enclosed. We added this information following line 171: "***The polynomial fit better explains long-term variability of sea level at Churchill compared to the linear approximation, with respective coefficients of determination ($R^2$) of 0.41 and 39***". *Thus, in our study we examined the sea level anomalies (SLA) against the low-frequency trend conditioned by the post-glacial isostatic adjustment* (lines 171-173).

10. Lines 172-173: It is not clear if the inverse barometer correction referred to here is a correction applied to the tide gauge record (in which case it should be clarified that the raw data represented measurements using an unvented tide gauge transducer) or simply a removal of the inverse barometer (IB) contribution to the water level record. If the latter, it would be worth commenting on the magnitude of the contribution of IB to interannual SLA, given studies indicating it is often non-negligible (e.g. Piecuch & Ponte 2015) and the potentially significant correlation with vorticity/wind effects.

We clarified our approach revising text in lines 172-173: "***In addition,*** *the inverse barometer contribution to the water level record was removed **using sea-level atmospheric pressure from the NCEP reanalysis**".* Following this sentence, we added information on the magnitude of correction attributed to IB: "***The mean correction attributed to inverted barometer effect was –1.19 ± 8.72 cm***". For sure, IB correction

is correlated to cyclonic wind and vorticity. Overall, correlation between vorticity and SLA was reduced by ~0.02-0.06 after removing IB. However, this contribution is not directly related to the ocean dynamics, and after discussing this point with co-authors it has been decided to remove the IB contribution.

11. Lines 299-301: There have likely been changes in sea ice cover over the reference period. How is this reflected in the analysis, if at all? Given the dominance of wind effects under ice-free conditions, perhaps there is value in discussing potential future changes if, as expected, ice cover continues to decline.

Thank you for this question. Since the ice-covered and ice-free periods in our analysis fixed to December-May and June-November, we cannot elaborate how changes in sea ice cover impact the results of our correlation analysis. *Andrews al.* (2017) revealed no trend for the open water period at Churchill for 1996-2016. For the entire Hudson Bay, however, the mean shifts toward a longer open water season (1980–1995 vs. 1996–2010) of ~3 weeks was revealed by *Hochheim and Barber* (2014). For pointing out potential future changes associated with ice cover decline, we added new sentence following line 456: "***In this context, transition towards a longer open water season (e.g., Hochheim and Barber, 2014) is expected to increase the contribution of atmospheric forcing to sea level variability***".

12. Line 308: Is it correct to say that sea level variability at Churchill is primarily impacted by wind forcing, when the authors state that wind forcing only explains ~22% of variability (Line 290)?

We modified this sentence to address concern by Reviewer #1 as follows: "*Our results show that sea level variability at Churchill is **rather influenced** by wind forcing…*"

13. Lines 310-312: Is there evidence to support the hypothesis that the dominant driver of storm surge on the western shore is Ekman transport, as opposed to a direct response of the water level to balance wind stress acting on the surface, or local bathymetric influences on storm surges? Although Figure 2b shows a strong correlation between the northerly wind component and vorticity, it would be helpful to more clearly explain whether these conditions are typically associated with an onshore (easterly) wind component or not, to confirm the relative importance of Ekman transport versus wind stress.

As pointed out by Reviewer #1, positive atmospheric vorticity over western Hudson Bay is associated with northerly winds (Figure 2b). Taking into account correlation between vorticity and SLA at Churchill, one may conclude that storm surge on the western shore is strongly impacted by Ekman transport. Moreover, there is no correlation between SLA and zonal wind (R = –0.03). The local bathymetry along the western shore is rather gentle, with depth gradients of ~2.5 m km$^{-1}$ (Figure 1). Following this comment by Reviewer #1, we added new sentence following line 313: "***A direct response of the water level to balance wind stress acting on the surface does not play a role for generating SLA because there is no correlation between SLA and zonal wind (not shown)***".

14. Lines 318-319: "It seems that these factors can explain the residual fraction of the SLA seasonal variability that is not explained by wind forcing and local river discharge." If I understand correctly, the authors concluded (Line 292) that wind effects and river discharge explain ~28% of SLA variability. The authors are therefore hypothesizing that the remainder, which at ~72% would represent a substantial (even dominant) contribution, is explained by thermosteric and halosteric effects. Is it then appropriate to refer to these as the "residual fraction"?

Following this comment, we changed "*residual fraction*" to "**significant** *fraction*".

15. Lines 357-360: The vorticity threshold described appears to imply that sea ice conditions (i.e., concentration, thickness, roughness, mobility as opposed to simply presence/absence) have little or no influence on the role of wind forcing in explaining SLA variability. I would expect ice conditions (e.g., presence or absence of shorefast ice and/or dynamic ice, roughness, concentration, fraction of Hudson Bay covered by ice) to play a role in determining the thresholds at which wind influences (positively or negatively) the SLA.

We agree with this comment. That is why we identify this threshold as "*a very rough estimate of the vorticity threshold attributed to the sea-ice impact*", lines 357-358. To address concern by Reviewer #1, we added new sentence following line 360: "***In fact, extension of the landfast ice as well as sea-ice roughness and concentration can play a role modifying the thresholds at which wind impacts the SLA***".

16. Lines 377-378 and 385: Which hypothesis by Gough and Robinson (2000) and Gough et al. (2005) are the authors referring to in this section? The preceding paragraph identifies multiple distinct hypotheses, and the linkage is not clear. Both paragraphs would benefit from more clearly identifying which of G&R (2000) and G et al (2005) hypotheses are debunked, and which are simply subject to uncertainty.

First, we modified these sentences to clarify which hypothesis we are referring to in this section: "*Overall, the hypothesis by Gough and Robinson (2000) and Gough et al. (2005)* **about the linkage between the river discharge pulse into James Bay and a positive SLA in Churchill** *is suggestive of the seasonal disruption of the Hudson Bay cyclonic circulation...*" and following line 385: "*In this context, the hypothesis by Gough and Robinson (2000) and Gough et al. (2005)* **linking SLA in Churchill linking SLA in Churchill to river discharge in James Bay** *seems to be inconsistent*".

Second, actually, there is only one hypothesis by *Gough and Robinson* (2000) and *Gough et al.* (2005) explaining a positive SLA observed in Churchill from October-November by the river discharge pulse into the James Bay region with an advective lag of ~4-5 months. This hypothesis was discussed and criticized in lines 377-387 and 400-424. We added new sentence summarizing the second portion of this critics following line 424: "***Overall, our third and fourth points suggest that the hypothesis of Gough and Robinson (2000) and Gough et al. (2005) about a linkage between river discharge into James Bay and SLA in Churchill is inconsistent***".

Third, *Gough and Robinson* (2000) and *Gough et al.* (2005) reported high correlations between Churchill River discharge and SLA in Churchill. We called these correlations into question in lines 388-399. We added new sentences to point this out following line 393: "***This calls into question the correlations between Churchill River discharge and SLA in Churchill reported by Gough and Robinson (2000) and Gough et al. (2005)***" and following line 399: "***This is consistent with a previous concern about significant impact of Churchill River discharge on SLA in Churchill***".

17. 1. Lines 417-418 (and 98-108): Figure 10 shows continuous contours of SSH derived from satellite altimetry data, which are discrete measurements at discrete intervals, usually along widely space satellite tracks.

The pattern shown in Figure 10 is used for qualitative purposes only. While the original altimetry measurements are along the satellite ground tracks, in this manuscript we used a gridded altimetry product. This product is obtained by optimally interpolating measurements from several (at least 2)

satellites. The procedure has been widely used since the beginning of 2000s (e.g., *Ducet et al.*, 2000, doi: 10.1029/2000JC900063). The details of the methodology can be found in *Pujol et al.* (2016, doi: 0.5194/os-12-1067-2016) and references therein.

17.2. Altimetry data is also notoriously uncertain near the land/water interface. It would be helpful to provide some additional context on the number of observations, spatial and temporal intervals, how the contours were developed, and the uncertainty associated with the SSH measurements (considering absolute differences shown are no more than +/- 5cm). To what extent is the satellite altimetry analysis influenced by spatial and temporal resolution of data and uncertainty in interpreted sea level measurements in proximity to the land/water interface?

Indeed, the quality of satellite altimetry data is compromised within 10-15 km offshore. Therefore, the gridded altimetry products usually discard radar measurements near the coast, for which specific processing is required. Note that disregarding the 10-15 km coastal band in Figure 10 does not change the general pattern and the interpretation of the result.

17.3. It would be helpful to provide some additional context on the number of observations, spatial and temporal intervals, how the contours were developed, and the uncertainty associated with the SSH measurements (considering absolute differences shown are no more than +/- 5cm). To what extent is the satellite altimetry analysis influenced by spatial and temporal resolution of data and uncertainty in interpreted sea level measurements in proximity to the land/water interface?

The gridded satellite altimetry data has been widely used for more than 2 decades. The mapping procedure has been described in many papers (see *Pujol et al.*, 2016 and references therein). While additional details on the procedure can be found in references, we do mention in the revised version of the manuscript (2nd paragraph of Section 2.1) that the root-mean-square differences between tide gauge records and collocated SLA/ADT data are 3-5 cm globally (e.g., *Volkov et al.*, 2007; *Pascual et al.*, 2009; *Volkov et al.*, 2012): "***The root-mean-square differences between tide gauge records and collocated SLA/ADT data are usually 3-5 cm (e.g., Volkov et al., 2007; Pascual et al., 2009; Volkov et al., 2012) and do not exceed 10 cm globally (CLS-DOS, 2016). When the altimetry data are averaged to produce the seasonal climatology, the measurement error is greatly reduced (at least by an order of magnitude for 28 years of altimetry record). It should be noted that altimetry errors near the coast are greater than in the open ocean. This is due to land contamination within the radar footprint and to the fact that the geophysical corrections applied to altimetry data are usually optimized for the open ocean and not for the coastal zones. In classical altimetry products, however, a large percentage of data within 10–15 km from the coast is deemed invalid and not used for generating SLA/ADT maps (e.g., The Climate Change Initiative Coastal Sea Level Team, 2020). Furthermore, satellite altimetry data was used here only for a qualitative assessment of the basin-scale seasonal sea-level patterns in Hudson Bay. Therefore, the reduced quality of altimetry retrievals near the coast is not expected to impact the conclusions of this study. Sea ice also does not represent a significant problem for computing the climatology, because Hudson Bay is essentially ice free during these months, especially during SON***". The actual accuracy of the altimetry data is even less than this range of values, because of the uncertainties in tide gauge records. Furthermore, when we compute the seasonal climatology, then the accuracy of a single altimetry measurement improves by at least an order of magnitude (given that there are 28 years and 9 independent measurements in a season). We also mention that sea-ice does

not represent a problem for computing the climatology, because Hudson Bay is essentially ice free during the months considered. Therefore, the pattern shown in Figure 10 is robust.

18. Lines 439-440: To what extent might anomalous inflows/outflows be controlled by the spatial distribution of sea ice (also potentially influenced by prevailing wind conditions)?

In this text referred by Reviewer #1 we make a reference to *Piecuch and Ponte* (2015). They found that sea level variability in Hudson Bay is driven by wind stress over Hudson Strait. We assume that in this context, anomalous inflows/outflows will be controlled by interplay between the spatial distribution of sea ice and wind forcing. However, given the lack of an interactive sea-ice model, *Piecuch and Ponte* (2015) did not simulate any role played by sea ice in mediating the transfer of momentum between the atmosphere and the ocean.

19. Lines 450-451: Odd use of "While…" to begin the sentence. Perhaps replace with "On the contrary".

We omitted "*While*" starting this sentence with "*Cyclonic atmospheric forcing…*".

20. Lines 466-470: I'm not sure the argument regarding the link between geostrophic flow and SLA is sufficiently put to rest without some analysis of the shore-perpendicular component of wind.

Following this concern by Reviewer #1, we lessened this statement to "*This seasonal pattern in sea-level variability **seems to have** implication for geostrophic circulation*".

21. Lines 760-765: Symbols (concentric circles and crosses) require legend or definition within caption.

We added this information in the Figure 11 caption: "***Dotted and crossed circles depict southerly and northerly along-shore surface winds, respectively***".

---

## Author Comment (AC2)

We highly appreciate helpful comments and suggestions by Reviewer #2. In the following, the comments by Reviewer #2 are underlined and our responses to the comments are in normal characters. Modifications to the text are shown in quotation marks with bold characters indicating newly added text, and normal characters indicating text that was already present in the previous version. The line numbering is referenced to the original manuscript version.

On behalf of the authors,

Igor Dmitrenko

**Reviewer #2**

The authors explore the relative contribution of dynamic atmospheric forcing and river discharge on variability of the sea level next to Churchill River. They also go beyond that and make some conclusions about the Hudson Bay.

Overall comments:

1. The paper will be a great contribution to the field after some minor issues are solved. I find the analysis convincing and the text well written. My comments mainly concerns better description of the datasets, complying with the data policy of the journal and FAIR scientific practice, as well as better representation of information on figures.

We appreciate this favorable evaluation of our manuscript by Reviewer #2. The issues related to description and presentation of the datasets along with other minor points were resolved following specific comments by Reviewer #2. Particularly, we added new section describing data availability.

2. Just one general note before the detailed comments. The authors jump from figure to figure in the text that makes it quite difficult to follow. I understand the necessity of coming back to the previous figures from time to time, but forcing the reader to do it so often is a bit cruel, in my opinion. I am not sure about the reasons to stick to the format when figures are inserted to the back of the manuscript. I think the OS format allows you to have them next to the text, which is much easier for reviewers.

We appreciate Reviewer #2 for pointing this out. For future submissions, we will move figures directly to the text to make them easier to follow.

**Detailed comments:**

3. 88-89 "This is the only permanently operating tide gauge in Hudson Bay and the central Canadian Arctic." - I think you already stressed this enough in the introduction, I would remove this repetition.

Omitted as recommended.

4. 98-108 There are two major points regarding satellite altimetry data that I miss:

4.1. There is no discussion of errors associated with the satellite measurements of sea level close to the coast, that is the main area of interest in this paper. It would be nice to at least mention it here, ideally provide the error estimates.

Reviewer #2 is correct saying that errors in satellite altimetry data near the coast are greater than in the open ocean. This is due to land contamination within the radar footprint and to the fact that some geophysical corrections applied to altimetry data are usually optimized for the open ocean and not for the coastal zones. In the revised manuscript (newly introduced 2nd paragraph of Section 2.1: "***The root-mean-square differences between tide gauge records and collocated SLA/ADT data are usually 3-5 cm (e.g., Volkov et al., 2007; Pascual et al., 2009; Volkov et al., 2012) and do not exceed 10 cm globally (CLS-DOS, 2016). When the altimetry data are averaged to produce the seasonal climatology, the measurement error is greatly reduced (at least by an order of magnitude for 28 years of altimetry record). It should be noted that altimetry errors near the coast are greater than in the open ocean. This is due to land contamination within the radar footprint and to the fact that the geophysical corrections applied to altimetry data are usually optimized for the open ocean and not for the coastal zones. In classical altimetry products, however, a large percentage of data within 10–15 km from the coast is deemed invalid and not used for generating SLA/ADT maps (e.g., The Climate Change Initiative Coastal Sea Level Team, 2020). Furthermore, satellite altimetry data was used here only for a qualitative assessment of the basin-scale seasonal sea-level patterns in Hudson Bay. Therefore, the reduced quality of altimetry retrievals near the coast is not expected to impact the conclusions of this study. Sea ice also does not represent a significant problem for computing the climatology, because Hudson Bay is essentially ice free during these months, especially during SON***"), we have mentioned this problem and explained that in classical altimetry products, including the one we are using, a large portion of invalid data within 10–15 km from the coast is not used for generating SLA/ADT maps. We also point out that we use altimetry data only for a qualitative assessment of basin-scale seasonal sea-level patterns. Therefore, the degraded quality of altimetry data within ~15 km from the coast does not represent a problem and does not change the conclusions. As requested by both reviewers, we reference studies that performed quality assessments of satellite altimetry data by comparing it to tide gauge records worldwide (*Volkov et al.*, 2007; *Pascual et al.*, 2009; *Volkov et al.*, 2012). As shown in these studies, the root-mean-square differences between tide gauge records and collocated SLA/ADT data are usually 3-5 cm and less than 10 cm (see also Validation of altimeter data by comparison with tide gauge measurements: yearly report 2016, https://www.aviso.altimetry.fr/fileadmin/documents/calval/validation_report/annual_report_TG_2016.pdf). Furthermore, in the revised manuscript, we also point out that averaging altimetry data over the two selected seasons reduces the measurement error at least by an order of magnitude.

4.2. It looks like there was substantial post processing involved in the altimetry data preparation for this paper. In order to comply with OS data policy (https://www.ocean-science.net/policies/data_policy.html) authors should provide the post-processed fields, and ideally the code that was used to generate them. If this is not possible, the explanation should be given in the data availability section. This is actually related to all data presented in the paper.

No substantial post processing of altimetry data was involved in this study. The daily SLA/ADT maps with all corrections applied are distributed via CMEMS (https://marine.copernicus.eu/). As described in the text, we only (i) computed the global mean sea-level, (ii) subtracted it from each grid point in Hudson Bay, and (iii) computed the seasonal climatology by averaging all available maps for each season. We added this statement in new section ***Data availability.***

5. 110-115 Again, please provide the extended time series, or explain why it's not possible.

The time series of the Churchill River discharge is added to supplementary material. They are referenced in lines 186-188 as follows: "*Churchill River discharge data were obtained from Déry et al. (2016) and extended to 2019; thus, we use a continuous record of daily mean discharge from 1960 to 2019 (Figure 4a **and supplementary material**)*". We also added this statement in new section **Data availability**.

6. 118 Please indicate where exactly the NCEP reanalysis was downloaded from.

We updated text with this information in lines 170-171: "*...were derived from the NCEP atmospheric reanalysis (**https://psl.noaa.gov/data/composites/hour/**...*". We also added this statement in new section **Data availability**.

7. 120 You use the ERA5 data after all (Fig.2), but this sentence makes me believe you disregard it. Please describe what ERA5 data was used, and where you have downloaded it from (or that you use it through Smith et al., 2014, as described below).

The reason for involving ERA5 is provided in lines 142-145: "*We also conducted a validation comparing the NCEP-derived vorticity to that derived from the ERA5 SLP utilizing the Web-Based Reanalysis Intercomparison Tools (**https://psl.noaa.gov/cgi-bin/data/testdap/timeseries.pl**) described by Smith et al. (2014). The comparison showed insignificant differences between the two reanalyses: the NCEP-derived vorticity only slightly exceeds that obtained from ERA5, while the correlation between the NCEP and ERA5-derived vorticities is 0.96 (Figure 2a)*". In this sentence we added the data source for the ERA5 data. We also added information on the ERA5 SLP data following line 122: "*However, we used the ERA5 SLP data to validate atmospheric vorticity derived from NCEP as described below in section 3*".

8. 122-124 Please explain the advantages of manual cyclone tracking over using one of the automatic tracking algorithms that use objective criteria for identifying cyclones. I am not saying it's a bad approach, just a word of justification would be nice.

We used cyclone manual tracking for simplicity. We added this information in section 2.3: "***For simplicity,** cyclones over the Hudson Bay area were manually tracked **for August-May 1969-1970 and 2003-2004***...". In this research, we used cyclone tracking only for August-May 1969-1970 and 2003-2004 (Figure 7). For these short time periods, it seems to be unnecessary to involve the automatic tracking algorithms. In the upcoming research focused on climatic forcing of the Hudson Bay circulation, we do this automatically following *Crawford et al.* (2021, doi: 10.1175/mwr-d-20-0417.1).

9. 145 I am not really sure what you are trying to tell with your Figure 2a. First, the correlation of 0.96 is not surprising, as you correlate two time series that have seasonal cycles, it says little on how one is similar to another, just that they have seasons. If you want to do the correlation, I would at least do the 12 month running mean on the time serieses before. But even then, what is the purpose of validating NCEP vs ERA5? What is the reference in this case? If you think ERA5 is better, why not just use it in the rest of the study, as the data are available? Showing that two reanalyses agree or disagree with each other without comparison to observational data does not make sense to me. Authors should better articulate the purpose of the comparison, or maybe just delete Figure 2a and respective text.

In general, ERA5 is better due to higher spatial resolution. However, "*We chose the NCEP reanalysis to extend the atmospheric forcing data back to 1950, which covers the tide gauge record from Churchill...*", lines 119-120. In this context, Figure 2a aimed to show that NCEP derived vorticity is very comparable to

vorticity obtained using ERA5. We added this motivation following line 122: "***However, we used the ERA5 SLP data to validate atmospheric vorticity derived from NCEP as described below in section 3***".

10. Figure 2. Caption says "NCEP and ERA5 (1970-2000)", but it looks like ERA5 data only starts in 1979.

Yes, Reviewer #2 is correct. We changed last sentence of Figure 2 caption to "*...the monthly mean vorticity derived from NCEP **(1949-2000)** and ERA5 **(1979-2000)** and (b) the monthly mean NCEP vorticity **versus** meridional wind (1949-2020)*".

11. Figure 3. The sentence "91-day running mean of daily atmospheric vorticity index (red, $s^{-1}$) over Hudson Bay and daily mean sea level measured at the tide gauge in Churchill (blue, m)." is confusing, and reads as if the vorticity index is smoothed and sea level is not. The time series continues to be very noisy, while below you nicely work with 365-day running mean, that filters out the seasonal cycle completely. Please consider showing 365-day running mean for vorticity index and sea level in the upper panel, while keeping the running correlation in the lower panel. I understand that your motivation for using 91-day running mean is to preserve the seasonal cycle to some extent, but this just does not work visually, we are looking at the noise.

The 91-day running mean was applied to both (i) the daily atmospheric vorticity and (ii) daily mean sea level measured at the tide gauge in Churchill. We modified the Figure 3 caption to make this clearer. As for the second part of this comment, we do not agree with Reviewer #2. As pointed out by Reviewer #2, our motivation for using 91-day running mean in Figure 3a was to preserve the seasonal cycle and its interannual variability. Figure 3b shows correlations computed between vorticity and SLA computed for the 365-day moving window. This correlation is mainly controlled by coherence of the seasonal cycles in vorticity and SLA shown in Figure 3a. In this context, gray shading highlighted the periods when correlations between vorticity and SLA was disrupted because there was no coherence between vorticity and SLA in Figure 3a. In contrast, yellow shading highlighted the periods when vorticity and SLA are correlated due to the coherent seasonal cycling in vorticity and SLA revealed in Figure 3a. That is why using 91-day running mean in Figure 3a is important for explaining patterns of correlations shown in Figure 3b.

12. Figure 4. Same comment as for Figure 3. I would still prefer to see a 365-day running mean, currently the upper panel does not convey any useful information to me, except that there is seasonal cycle in both discharge and SLA, but this is not worth a figure. The change after diversion will still be visible on the 365-day running mean.

Similar to comment #11 by Reviewer #2, the 91-day running mean in Figure 4a is intended to show the seasonal cycling and its interannual variability. Figure 4a provides physical background for explaining correlations in Figure 4b.

13. 186-189 Figure 5 is a much better illustration of seasonal cycle changes than Fig. 3 and 4a.

The text referenced by Reviewer #2 is "*The long-term variability of sea level (Figure 3a) and SLA (Figure 4a) shows no abrupt disruption with the introduction of the Churchill River diversion in 1977. However, the seasonal cycle of SLA generated for pre- and post-diversion shows a characteristic difference in the timing and magnitude of SLA (Figure 5a)*". Here Figures 3a and 4a intended to show no abrupt disruption in sea level associated with the introduction of the Churchill River diversion in 1977. However, Figure 5 definitely shows changes in the seasonal cycle. This is exactly in line with this comment by Reviewer #2.

However, Figure 5 does not show the interannual variability of seasonal cycle. This information can only be derived from Figures 3a and 4a.

14. 221-224 I was trying to find on the very noisy Figure 3a what you are talking about, but failed desperately. I understand that for the person who looks at these graphs long enough there is no problem to distinguish between "late fall and beginning of winter", but for the mere mortal that just sees these graphs for the first time it's just too much. Please, either highlight the periods you are talking about, or just find some other way to demonstrate them.

We agree with this critics by Reviewer #2. In the revised version of our manuscript, we indicated these periods of diminished and enhanced seasonal vorticity in Figure 3a by black and green triangles, respectively. The corresponding information was added in lines 223 and 225. New sentence was also added to Figure 3 caption: "***Black and green triangles show periods when seasonal vorticity from late fall to early winter was diminished and amplified, respectively***".

15. 227-231 I am sorry, but you can't expect the reader to identify October-November on Figure 3a.

In this context, reference to October-November 1969 and 2003 was associated with Figure 6, not Figure 3a. In Figure 3a, the entire periods from August 1969/2003 to May 1970/2004 were highlighted with yellow shading. In Figure 6, these periods were enlarged.

16. Figure 6 running mean lines are almost invisible, please make them thicker, or use more contrast colors.

We made these lines in Figure 6 thicker.

17. 314-321 It is unfortunate that the authors decide not to include analysis of thermo- and halosteric effects, as they might show interesting interplay between atmospheric forcing and ocean thermodynamics. I can understand that it might be too much for one paper, but it would be nice if the authors return to it in the future work.

We appreciate Reviewer #2 for this comment. Analysis of thermo- and halosteric effects requires numerical simulations. While Baysys project includes NEMO simulations (U of Alberta, group led by Paul Myers), they were not used for this specific research. This would be a separate topic for future research.

18. 443-445 Can you make any speculation on how possible changes in cyclone activity due to climate change may affect the sea level variability in the Bay? Or maybe 471-475 is a better place for it.

As follows from Figure 3a, atmospheric vorticity over Hudson Bay does not show a trend that can be attributed to climate change. Moreover, while working on the manuscript focused on climate forcing of the Hudson Bay circulation, we revealed teleconnections between vorticity and El Niño and La Niña events. That is why the issue pointed out by Reviewer #2 is not clearly understood, and at the moment, we would like to avoid speculations on this subject. Following this comment, we introduced new sentence to finalize our summary and concussions: "***Possible impacts of climate change on cyclone activity in Hudson Bay, and therefore sea-level variability, will be addressed in future research***".

---

## Author Response (AR2)

Dear Dr. J. Williams,

We highly appreciate your efforts for evaluating our manuscript as well as for your final comments and suggestions. In the following, your comments are underlined and our responses to the comments are in normal characters. The line numbering is referenced to the marked-up manuscript version.

On behalf of the authors,

Igor A. Dmitrenko

Dear authors,

Thank-you for the resubmission and response to the reviewers. I am satisfied that you have addressed their comments, and for the paper to proceed without going back out for re-review. I have picked up a few small points to address, please see below. Line numbers refer to version 3 of the manuscript:

1. Line 163 JJA - do you mean June through August? Try to be consistent about naming of these periods.

Yes, we mean June through August. Text in line 164 was corrected accordingly.

2. Line 164 and 179 You could cut the repetition here.

We omitted this repetition in lines 180-182.

3. Table captions. The repeated "and" is ambiguous. I suggest eg "Correlations (R) of monthly-mean atmospheric vorticity and/or Churchill River discharge against sea level anomalies in western Hudson Bay"

Changed as recommended, lines 906, 910, and 914.

4. I think you're right to use daily means of the SLA, that's fine. But as far I can see you took daily means of the hourly tide gauge data, but there's no mention of a tidal filter (eg Doodson filter). If you just take daily means of hourly data without filtering, aliasing of the M2 tide will leave a fortnightly cycle with amplitude about 0.03 of M2 amplitude, ie about 4cm. I don't think it will hugely affect your results, but it might reduce correlation slightly, and would show up in figure 6. Please clarify whether you did this.

Sea level data were de-tided using an algorithm by *Foreman* (1977). We added this information in lines 137-138.

5. It is still hard to see all the lines in figure 6, especially the period of most interest where the spikes coincide. It is possible if I zoom in so I won't absolutely insist you replot it, but it would be better if you do. I suggest a darker shade of blue, to lighten the yellow highlighting and to ensure the highlight is the bottom layer. It would also help a lot to reduce the aspect ratio of the plot (make it less tall but the same width) as this reduces the spikiness of the peaks and makes them clearer.

We modified blue line in Figure 6 with a darker shade of blue as recommended. The yellow highlight has been moved to the bottom layer.

6. I like the way you've used the line colour on the axes as a legend.

Thank you.

7. The discussion is quite long. I suggest breaking it up with some subheadings to aid the reader.

We agreed with this comment. Actually there are three contributors to SLA assessed in discussion: wind forcing, river discharge forcing and thermohaline (thermosteric and halosteric) forcing. The problem, however, is that there is interplay between these three contributors, and discussion is written in a way that they are cannot be easily separated without substantial reorganizing of discussion. Respecting this comment by the Editor, we would like to keep the existing structure of discussion.

8. Line 547 and elsewhere - remove apostrophe after time series.

Corrected in lines 477 and 548.

9. The SLA scale is wrong on figure 8.

Thank you, fixed.

10. The paragraph about resupply starting 663 seems a bit redundant. line 664 "highly rely" -> rely heavily

In abstract, we pointed out the importance of sea level variability for the coastal infrastructure in lines 33-34: "*Therefore, understanding sea-level variability at the Port is an urgent issue crucial for safe navigation and coastal infrastructure*". This paragraph in lines 665-670 highlights this point providing additional context of our research.  We modified text in line 665 as recommended.

11. As you finalise the paper, you may like to consider a graphical and/or video abstract to explain your paper to a wider audience. This doesn't have to be ready at the same time as publication, but do think about what you'd put in this.

We appreciate your advice.